# 🦬 MAGNET: A Multi-agent Framework for Finding Audio-Visual Needles by Reasoning over Multi-Video Haystacks

**Sanjoy Chowdhury**[1], **Mohamed Elmoghany**[2], **Yohan Abeysinghe**[3], **Junjie Fei**[2], **Sayan Nag**[4],
**Salman Khan**[3], **Mohamed Elhoseiny**[2], **Dinesh Manocha**[1]

[1]University of Maryland, College Park    [2]KAUST    [3]MBZUAI    [4]University of Toronto

{sanjoyc, dmanocha}@umd.edu    sayan.nag@mail.utoronto.ca    mohamed.elhoseiny@kaust.edu.sa

{yohan.abeysinghe, salman.khan}@mbzuai.ac.ae    m.osama.elmoghany@gmail.com

🌐 https://schowdhury671.github.io/magnet_project/

## Abstract

Large multimodal models (LMMs) have shown remarkable progress in audio-visual understanding, yet they struggle with real-world scenarios that require complex reasoning across extensive video collections. Existing benchmarks for video question answering remain limited in scope, typically involving one clip per query, which falls short of representing the challenges of large-scale, audio-visual retrieval and reasoning encountered in practical applications. To bridge this gap, we introduce a **novel task** named **AVHaystacksQA**, where the goal is to identify salient segments across different videos in response to a query and link them together to generate the most informative answer. To this end, we present **AVHaystacks**, an audio-visual benchmark comprising 3100 annotated QA pairs designed to assess the capabilities of LMMs in multi-video retrieval and temporal grounding task. Additionally, we propose a model-agnostic, multi-agent framework **MAGNET** to address this challenge, achieving up to 89% and 65% relative improvements over baseline methods on BLEU@4 and GPT evaluation scores in QA task on our proposed AVHaystacks. To enable robust evaluation of multi-video retrieval and temporal grounding for optimal response generation, we introduce two new metrics, **STEM**, which captures alignment errors between a ground truth and a predicted step sequence and **MTGS**, to facilitate balanced and interpretable evaluation of segment-level grounding performance.

## 1 Introduction

Large Multimodal Models (LMMs) [1–6] have achieved remarkable progress in audio-visual understanding. However, they continue to face significant challenges [7] when it comes to retrieving and reasoning over large-scale multimedia collections particularly in tasks such as audio-visual retrieval-augmented generation (RAG) and multi-video temporal grounding. These limitations hinder their effectiveness in real-world applications, such as querying personal video archives, how-to repositories, or educational video libraries, where complex queries often require joint processing of both audio and visual modalities across multiple video segments.

Despite their growing capabilities, to the best of our knowledge, no existing task or benchmark systematically evaluates LMMs' ability to identify and integrate salient segments from multiple videos to construct informative, grounded responses. As a result, there remains a gap in properly assessing their performance on large-scale audio-visual retrieval and reasoning tasks. Existing benchmarks [8–11] are generally limited in scope typically associating each question with only a single short clip, as shown in Tab. 1. However, real-world information-seeking scenarios often

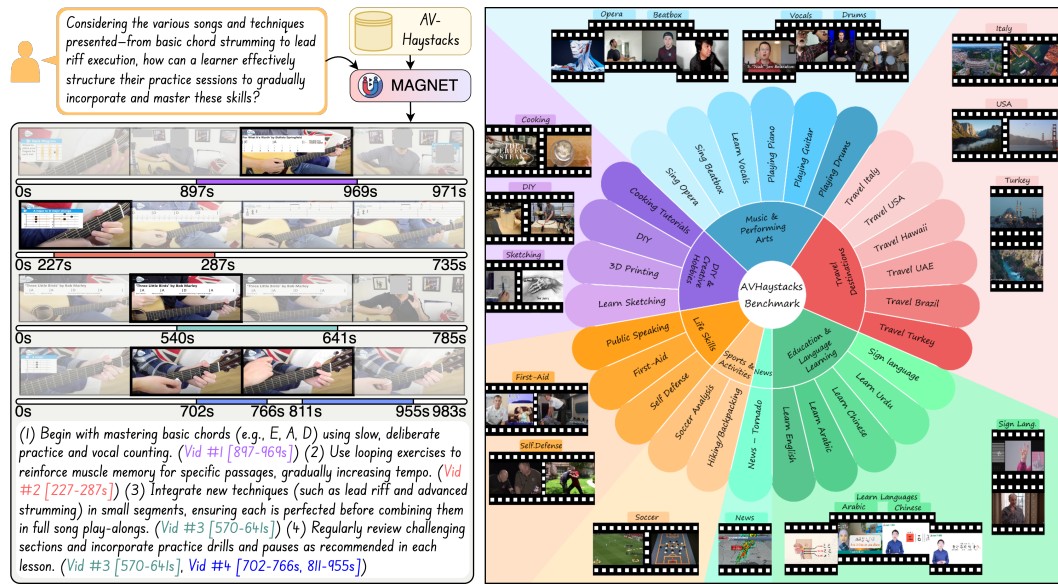

Figure 1: **A new task and a benchmark**. We introduce a novel task AVHaystacksQA, which requires multi-video linkage and reasoning to generate the most informative answer. To this end, we curate a new benchmark AVHaystacks comprising 3100 QA pairs obtained through careful inspection.

demand searching through hundreds or thousands of video segments, identifying the most relevant audio-visual snippets, and synthesizing coherent, evidence-backed answers from them.

To address this gap, we introduce a new benchmark **AVHaystacks** to facilitate novel task **AVHaystacksQA**. This task is designed to comprehensively evaluate the LMMs ability to perform large-scale, multi-video audio-visual retrieval and reasoning. Each query in our benchmark is grounded in a massive collection of up to **500** video clips, requiring models to localise relevant segments both temporally and across sources and reason over their combined audio and visual signals. This setup reflects the complexity of real-world information needs more accurately than previous single-video setups.

A central challenge in building such a benchmark is designing specific, unambiguous questions that genuinely require both audio and visual understanding and cannot be accurately answered using a single modality or video clip alone. To address this, we implement a robust data filtering pipeline that combines the strengths of large language models (LLMs) and human annotators to remove generic, redundant, or overly broad queries. For instance, questions such as *"How do I improve my strumming while playing guitar?"* or *"How do you adjust for tightness in your lips to hinder a clear operatic tone?"* are carefully selected and validated to ensure they demand cross-modal reasoning and temporal grounding within specific video segments.

To enable LMMs to tackle this challenging task, we introduce **MAGNET** a novel retrieval-augmented, multi-agent framework designed to find audio-visual needles within multi-video haystacks. Our approach integrates multiple specialized audio and video encoders to capture rich multi-modal semantics, and incorporates a multi-agent framework that scores the relevance of retrieved video segments with respect to the input query. By tightly coupling retrieval with reasoning, MAGNET facilitates efficient and scalable exploration of large-scale video corpora. Extensive experiments demonstrate that our framework significantly enhances both retrieval accuracy and answer generation performance.

**Main Contributions:**

**(1)** We **propose a novel task**, AVHaystacksQA, and introduce AVHaystacks a new benchmark consisting of 3100 audio-visual QA pairs drawn from videos across diverse domains (Fig. 1). This benchmark pushes the boundaries of video retrieval and reasoning by requiring models to navigate and reason over large-scale video collections. To the best of our knowledge, no existing benchmark systematically evaluates multi-video keypoint detection and reasoning capabilities.

| Dataset | Train | Test | MS | TA | MVL | AVR | AVD | RQA | AVQA | LC | Avg. Dur. (s) |
|---|---|---|---|---|---|---|---|---|---|---|---|
| *Video Datasets* | | | | | | | | | | | |
| ShareGPT4Video [12] | ✓ | ✗ | ✓ | ✗ | ✗ | ✗ | ✗ | ✗ | ✗ | ✗ | 26 |
| Cinepile [13] | ✓ | ✓ | ✓ | ✓ | ✗ | ✗ | ✗ | ✗ | ✗ | ✗ | 160 |
| NExT-QA [14] | ✓ | ✓ | ✗ | ✗ | ✗ | ✗ | ✗ | ✗ | ✗ | ✗ | 48 |
| Video-MME [9] | ✗ | ✓ | ✓ | ✗ | ✗ | ✗ | ✗ | ✗ | ✗ | ✓ | 1020 |
| LongVideoBench [10] | ✗ | ✓ | ✓ | ✓ | ✗ | ✗ | ✗ | ✗ | ✗ | ✓ | 480 |
| MovieChat [15] | ✓ | ✓ | ✗ | ✗ | ✗ | ✗ | ✗ | ✗ | ✗ | ✓ | 420 |
| *Audio-Visual Datasets* | | | | | | | | | | | |
| UnAV-100 [16] | ✓ | ✓ | ✗ | ✓ | ✗ | ✗ | ✓ | ✗ | ✓ | ✗ | 42 |
| VAST-27M [17] | ✓ | ✓ | ✗ | ✗ | ✗ | ✗ | ✓ | ✗ | ✓ | ✗ | 20 |
| AVQA [18] | ✓ | ✓ | ✗ | ✓ | ✗ | ✗ | ✗ | ✗ | ✓ | ✗ | 60 |
| AVInstruct [19] | ✓ | ✗ | ✗ | ✓ | ✗ | ✗ | ✗ | ✗ | ✓ | ✗ | 115 |
| Music-AVQA [20] | ✓ | ✗ | ✗ | ✗ | ✗ | ✗ | ✗ | ✗ | ✓ | ✗ | 10 |
| VGGSound [21] | ✓ | ✗ | ✗ | ✗ | ✗ | ✗ | ✗ | ✗ | ✓ | ✗ | 10 |
| AVBench+SAVEnVid [22] | ✓ | ✓ | ✓ | ✓ | ✗ | ✗ | ✓ | ✗ | ✓ | ✗ | 182 |
| **AVHaystacks (Ours)** | ✓ | ✓ | ✓ | ✓ | ✓ | ✓ | ✓ | ✓ | ✓ | ✓ | **738** |

Table 1: **Comparison with prior video/audio-visual benchmarks.** MS: Model-Assisted; TA: Temporal Annotation; MVL: Multi-Video Linkage; AVR: Audio-Visual fine-grained Reasoning; AVD: Audio-Visual Description; RQA: Retrieval-based QA Answering; AVQA: Audio-Visual QA; LC: Long Context, where QA context spans over 5 mins.

**(2)** We **conduct extensive evaluations** of state-of-the-art audio-visual models on AVHaystacks, analyzing their performance across a range of multi-video retrieval and reasoning setups. Our findings reveal that current models perform suboptimally in retrieving relevant videos from large corpora and struggle to reason effectively across multiple clips to identify the key segments needed to answer complex queries.

**(3)** To enable robust evaluation of audio-visual retrieval and grounded temporal reasoning, we **introduce two novel metrics**: STEM, which quantifies alignment errors between the ground-truth and predicted step sequences in multi-video audio-visual answer generation; and **MTGS**, which provides a balanced and interpretable assessment of segment-level grounding performance.

**(4)** Finally, we propose a **model-agnostic, multi-agent training strategy**, MAGNET, designed to enhance model performance in identifying key segments across multi-video haystacks. Experimental results show that our framework achieves up to 89% and 65% relative improvements over baseline methods on BLEU@4 and GPT-based evaluation scores, respectively, on AVHaystacksQA.

## 2 AVHaystacks: Audio-Visual Benchmark for Multi-Video Temporal Grounding and Reasoning

The benchmark curation pipeline consists of five stages: (1) *Video curation:* Following careful manual inspection, we collect 500 videos from YouTube spanning 27 diverse categories, including how-to guides, cooking, travel, musical instrument tutorials, language instruction, and vocal lessons. Each video is selected to ensure its suitability for audio-visual QA tasks specifically, queries where answering correctly requires a strong understanding of both audio and visual modalities, with complementary cues essential for accurate reasoning. (2) *Blind question generation:* Using OpenAI O3-MINI with custom prompts (details in the supplementary), we generate 50 topic-agnostic questions per topic. This promotes comprehensive evaluation by introducing diversity in reasoning types, modality dependence, and task complexity. (3) *Transcript cleaning and segmentation:* Subtitle overlaps are resolved, and transcripts are segmented into coherent instructional subtopics to support fine-grained QA generation. (4) *Segment-aware QA prompting:* Segment-specific questions are automatically generated to require multimodal comprehension leveraging audio, visual, and textual cues. (5) *Answer grounding:* Answers are constructed in a step-wise manner, referencing at least two distinct video segments. Additional details are provided in the supplementary material.

**Dataset Selection.** We apply four filtering criteria for video curation: (1) synchronized audio, on-screen text, and visual changes; (2) clear procedural step sequences; (3) duration between 5–25 minutes; and (4) availability of English captions.

**Grounded Audio Visual Question Answering.** Each QA item includes: (i) a free-form question, (ii) a step-by-step answer, and (iii) a list of ⟨videoID, start, end⟩ references. Unlike prior single-clip datasets, 82% of our QA pairs require evidence from at least two distinct videos, making them well-suited for LMMs. Examples of extended answers are shown in supplementary.

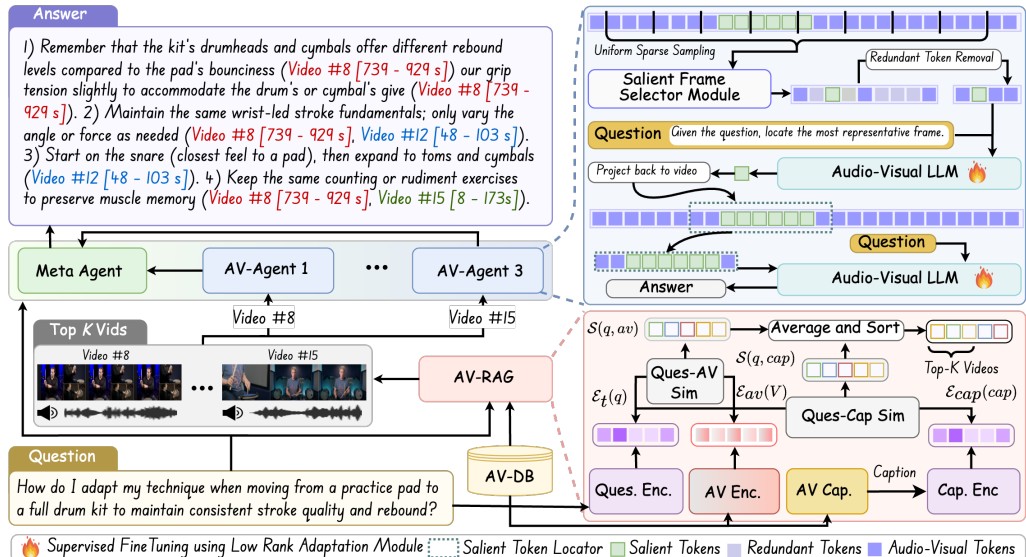

Figure 2: **Overview of MAGNET.** Given a user query, AV-RAG retrieves the top-K relevant videos (with audio), which are then processed by dynamically spawned Audio-Visual agents and a meta-agent aggregator to generate temporally grounded, step-wise responses. An adaptive, modality-agnostic frame selection module improves performance over baselines (see Tab. 2).

**Multiple Audio-Visual Entity Linkage.** To enable fine-grained, cross-video entity grounding without manual bounding boxes, we generated **3100** QA queries (train/test split 2k/1k) drawing evidence from a **500-video** pool. To facilitate experiments with baseline approaches we curate a small subset AVHaystacks-50. Instead of box-level labels, each entity is grounded via a sequence of multimodal spans: (i) textual segments from cleaned transcripts, (ii) audio intervals capturing distinctive sounds, and (iii) visual frame intervals showing the entity onscreen. These spans often span multiple videos in a defined order (e.g., steps 1–3 of a recipe across two clips), capturing both temporal and cross-video dependencies (details in supplementary).

## 3 Method

### 3.1 Task Definition: AVHaystacksQA

Given a question $q$ and a collection of $N$ videos $\mathcal{V} = \{V_1, \ldots, V_N\}$, our framework aims to retrieve the top-$k$ most relevant videos to support AVLLMs understanding and answering the question $q$. MAGNET accomplishes this through a *two-step* retrieval process designed to effectively identify and reason over relevant videos for each question as demonstrated in Fig.2.

### 3.2 Audio Visual Preprocessing

**AV RAG.** We compute similarity scores between the averaged query representation $q$ and (*i*) fused audio-visual features (using Hadamard fusion) $\mathcal{E}_{av}(\cdot)$ and (*ii*) encoded audio-visual captions $\mathcal{E}_{cap}(\cdot)$ for each video $V_j \in \mathcal{V}$ using cosine similarity, as defined in Eq. 1:

$$\mathcal{S}(q, \mathcal{V}) = \cos(\mathcal{E}_t(q), \mathcal{E}_f(V_j)) \quad \text{for} \quad V_j \in \mathcal{V}, \tag{1}$$

Here, $\mathcal{S}$ denotes the similarity score between query $q$ and video set $\mathcal{V}$; cos is cosine similarity; $\mathcal{E}_t$ is the text encoder; and $\mathcal{E}_f \in \{\mathcal{E}_{av}, \mathcal{E}_{cap}\}$.

We use IMAGEBIND [23] to encode both the query and audio-visual features. Captions are generated using Gemini 1.5 Pro [24] and encoded with IMAGEBIND to obtain $\mathcal{E}_{cap}(\cdot)$; all embeddings are cached in our retrieval database to enable fast similarity computation without runtime re-encoding. To compute final relevance, we average $\mathcal{S}(av, q)$ and $\mathcal{S}(cap, q)$ to obtain $\text{Sim}_{\text{avg}}$, then rank videos in descending order and select the top-$k$ most relevant videos.

**Salient Frame Selector Module (SFS).** Long videos often contain sparse but crucial moments relevant to a query. To efficiently localize these key events, we introduce a *salient frame selection*

**Algorithm 1** SFS

**Input:** $m$ total frames, target count $k$, matrix $Q$
**Output:** Selected frame indices
1: Initialize: $C[0\ldots m][0\ldots k] \leftarrow \infty, C[0][0] \leftarrow 0$
2: Initialize: $backtrack[0\ldots m][0\ldots k] \leftarrow -1$
3: **for** $j \in \{1,\ldots,k\}$ **do**
4:    **for** $i \in \{j\ldots m\}$ **do**
5:       **for** $p \in \{j-1\ldots i-1\}$ **do**
6:          **if** $C[p][j-1] + Q[p][i] < C[i][j]$ **then**
7:             $C[i][j] \leftarrow C[p][j-1] + Q[p][i]$
8:             $backtrack[i][j] \leftarrow p$
9: Initialize: $result \leftarrow [\ ], j \leftarrow k, i \leftarrow m$
10: **while** $j > 0$ **do**
11:    $result.\text{append}(i)$
12:    $i \leftarrow backtrack[i][j], j \leftarrow j-1$
13: **return** $result.\text{reverse}()$

---

**Algorithm 2** STEM: **St**ep-wise **E**rror **M**etric

**Input:** Ground Truth Steps: $\{G_1,\ldots,G_n\}$, Predicted Steps: $\{P_1,\ldots,P_m\}$, Text Similarity Threshold: $\tau_s = 0.5$.
**Output:** Missing Step: $S_M$, Hallucinated Step: $S_H$, Wrong Step Order: $S_O$, Step wise Video ID False Positives and Negatives: $S_{FP}, S_{FN}$, Step-wise IoU on time intervals: $S_{IoU}$, Similarity Matrix: $M_{sim}$, Step Similarity Function: $\text{Sim}(\cdot)$, Hungarian Matching Algorithm: $\text{Hung}(\cdot)$, Matched Steps: $\hat{G}T, \hat{P}$
1: $M_{sim} \leftarrow \text{Sim}(G_i^{\text{text}}, P_j^{\text{text}})$    ▷ Compute similarity matrix
2: $\hat{G}, \hat{P} \leftarrow \text{Hung}(M_{sim}, \tau_s, G, P)$    ▷ Obtain matched pairs
3: **for** matched pairs $(\hat{G}_i, \hat{P}_j)$ **do**
4:    **if** $i \neq j$ **then**
5:       $S_O \leftarrow S_O + 1$    ▷ Wrong Step Order
6:    **for** groundings $(v_{\text{pred}}, t_{\text{start}}^{\text{pred}}, t_{\text{end}}^{\text{pred}})$ in $P_j$ **do**
7:       **if** $v_{\text{pred}} \notin \{v_{gt} \in G_i\}$ **then**
8:          $S_{FP} \leftarrow S_{FP} + 1$    ▷ Video ID Mismatch
9:       **else**
10:          $S_{IoU} \leftarrow \text{IoU}\left([t_{\text{start}}^{gt}, t_{\text{end}}^{gt}], [t_{\text{start}}^{\text{pred}}, t_{\text{end}}^{\text{pred}}]\right)$
11:    **for** groundings $(v_{gt}, t_{\text{start}}^{gt}, t_{\text{end}}^{gt})$ in $G_i$ **do**
12:       **if** $v_{gt} \notin \{v_{\text{pred}} \in P_j\}$ **then**
13:          $S_{FN} \leftarrow S_{FN} + 1$    ▷ Video ID Mismatch
14: **for** unmatched $(G - \hat{G})_i$ **do**
15:    $S_M \leftarrow S_M + 1$    ▷ Missing Step
16: **for** unmatched $(P - \hat{P})_j$ **do**
17:    $S_H \leftarrow S_H + 1$    ▷ Hallucinated Step

---

module that focuses on visually and semantically important content. This ensures attention is directed to both the *right content* and the *right time*, facilitating efficient reasoning.

We begin by uniformly sampling $m$ candidate frames without any prior. The objective is to select $k$ representative frames that are both visually diverse and temporally dispersed.

Let $I_t$ denote the $t$-th sampled frame, and $z_t \in \mathbb{R}^d$ its (Hadamard) fused audio-visual embedding from ImageBind. We compute the pairwise cosine similarity between all frame pairs:

$$\Gamma_{ab} = \frac{z_a^\top z_b}{\|z_a\|_2 \cdot \|z_b\|_2}, \quad \forall a, b \in \{1, \ldots, m\} \tag{2}$$

To discourage temporally adjacent selections, we apply a temporal separation penalty to frame pairs, where $\gamma$ is the separation penalty factor:

$$\Delta_{ab} = \gamma \left( \frac{1}{\sin\left(\frac{\pi}{2}|a - b|\right) + 1} - 1 \right) \tag{3}$$

The total affinity matrix is defined as $\mathbf{Q}_{ab} = \Gamma_{ab} + \Delta_{ab}$. We then select a sequence of $k$ frame indices $\mathcal{T} = \{t_1, t_2, \ldots, t_k\}$ such that $1 \leq t_1 < \ldots < t_k \leq m$ and the total pairwise similarity is minimized (process detailed in Algorithm 1) using the following equation:

$$\mathcal{T} = \arg\min_{\substack{\mathcal{T} \subset \{1,\ldots,m\} \\ |\mathcal{T}|=k}} \sum_{i=1}^{k-1} Q_{t_i t_{i+1}} \tag{4}$$

### 3.3 Grounded Question-Answering with Audio-Visual agents

Once the potentially relevant videos are shortlisted (i.e., those likely to be helpful in answering the given query), we deploy a dynamic agentic setup based on an AVLLM backbone. Due to its recent success in fine-grained audio-visual comprehension, we utilise Qwen 2.5 Omni [25] as the backbone for our AVLLM agents. For each shortlisted video, a dedicated instance of Qwen 2.5 Omni is spawned to process that video independently. Each agent analyses its assigned video and predicts the most relevant temporal segments along with a response that may contain an answer to the query. The outputs from all individual AVLLM agents, the identified time windows and corresponding partial responses are then aggregated by a meta-agent. In our setup, GPT-4o[26] acts as the meta-agent,

which ingests the agent responses and synthesises a coherent, contextually grounded final answer for the input query.

# 4 Experiments

## 4.1 Metrics
We utilize a set of 4 metrics to evaluate the outcomes of our approach which are as follows:

**Response Alignment Scores.** We evaluate the semantic alignment between the summarized outputs, comprising step-wise responses to a given query, and the corresponding ground truth using a combination of automated and human-centric metrics. For automated evaluation, we employ standard text-based similarity metrics such as BLEU@4 and CIDEr. Additionally, we utilize the GTE-L model [27] to extract sentence embeddings for both the predicted response and the ground truth, and compute their cosine similarity. Beyond these, we adopt the GPT-as-a-Judge framework to score the predicted responses against the ground truth on a 10-point scale, subsequently normalizing these scores for consistency. Finally, we include human evaluation scores (on a scale 1-5 with 1 being lowest) as a complementary metric (averaged across 20 evaluators) to assess the alignment and overall quality of the predicted answers with respect to the ground truth.

**Retrieval Evaluation Scores.** To evaluate how well our system retrieves relevant videos from the haystacks, we present recall values using *R@1*, *R@3*, and *R@5*. These metrics help us assess the video retrieval accuracy by examining the presence of relevant videos within the top few ranks.

**Matched Temporal Grounding Score.** To evaluate the alignment between predicted and ground truth temporal segments (along with video IDs), for each query, we propose the *Matched Temporal Grounding Score* (MTGS). This metric measures the average temporal overlap (IoU) between predicted and ground truth time intervals, but only for video instances where the video IDs match. If a video ID is not present in both prediction and ground truth, it is excluded from the computation. Furthermore, for each matched video ID, we compute the temporal IoU over the union of all corresponding intervals. This ensures that the metric reflects segment-wise grounding accuracy at the video level. Formally, let $\mathcal{V}_G$ and $\mathcal{V}_P$ denote the sets of video IDs present in the ground truth and predicted outputs, respectively. Let $\mathcal{V}_M = \mathcal{V}_G \cap \mathcal{V}_P$ be the set of matched video IDs. For each $v \in \mathcal{V}_M$, let $G_v = \{(t_{\text{start}}^{\text{gt}}, t_{\text{end}}^{\text{gt}})\}$ be the set of ground truth intervals and $P_v = \{(t_{\text{start}}^{\text{pred}}, t_{\text{end}}^{\text{pred}})\}$ be the set of predicted intervals. We define the temporal IoU for video $v$ as: $\text{IoU}_v = \frac{\text{Duration}(\text{Intersection}(G_v, P_v))}{\text{Duration}(\text{Union}(G_v, P_v))}$, where the intersection and union are computed by merging overlapping intervals across $G_v$ and $P_v$, and $\text{Duration}(\cdot)$ computes the total length of the resulting intervals. The final MTGS is then computed as the mean IoU across all matched video IDs: $\text{MTGS} = \frac{1}{|\mathcal{V}_M|} \sum_{v \in \mathcal{V}_M} \text{IoU}_v$. In cases where there is no matched video ID between prediction and ground truth (i.e., $|\mathcal{V}_M| = 0$), we define $\text{MTGS} = 0$. This metric provides a balanced and interpretable evaluation of segment-level grounding performance, with sensitivity to both partial and full overlaps, while ensuring fairness by averaging over matched video contexts. We report the average value of this score $\text{MTGS}_{\text{avg}}$ in Tab. 3.

**STep-wise Error Metric.** The Step-wise Error Metric (STEM) quantifies alignment errors between a ground truth step sequence and a predicted step sequence in instructional or procedural data (detailed in Algorithm 2). Given Ground truth steps: $\{G_1, G_2, \ldots, G_n\}$, Predicted steps: $\{P_1, P_2, \ldots, P_m\}$, Text similarity threshold: $\tau_s \in [0, 1]$, typically set to 0.5, we begin by computing a similarity matrix $M_{\text{sim}} \in \mathbb{R}^{n \times m}$, where each entry is given by a cosine similarity function $\text{Sim}(G_i^{\text{text}}, P_j^{\text{text}})$ computed between the step-wise text embeddings. We obtain a set of valid matched pairs of predicted and ground truth steps via Hungarian matching [28] which minimizes the total dissimilarity. If $i \neq j$, the prediction is out of order, contributing to the wrong step order count. Subsequently, to assess grounding mismatch, for each matched step, we compare video IDs and compute corresponding IoUs between prediction and ground truth. Finally, unmatched ground truth steps add to the missing step count, and unmatched predicted steps are considered as hallucinated steps.

## 4.2 Baselines
In our experiment, we have evaluated several open and closed-sourced Audio-Visual models on the retrieval and AVHaystacksQA performance. We extensively evaluate AVHaystacks on VideoRAG [8], Video-RAG [29], Qwen 2.5 Omni [25], Unified IO2 [4], Video-SALMONN [30]. We suitably adopt Video-RAG, VideoRAG for our task. To accommodate videos in Qwen-2.5-Omni, Unified-IO2, Video-SALMONN for AVHaystacks-50 we sparsely sample frames and downsize them to low resolution and also compress audio using [31].

| | AVHaystacks-50 | | | | | AVHaystacks-Full | | | | |
|---|---|---|---|---|---|---|---|---|---|---|
| Method | B@4 ↑ | Cr ↑ | Text Sim ↑ | GPT Eval ↑ | H Eval ↑ | B@4 ↑ | Cr ↑ | Text Sim ↑ | GPT Eval ↑ | H Eval ↑ |
| VideoRAG [8] | 43.16 | 119.78 | 5.31 | 6.32 | 3.42 | 41.59 | 115.97 | 5.15 | 6.13 | 3.32 |
| Video-RAG [29] | 42.64 | 117.86 | 5.23 | 6.20 | 3.37 | 40.67 | 112.12 | 4.99 | 5.97 | 3.23 |
| Qwen2.5 omni [25] | 10.84 | 28.59 | 1.90 | 2.11 | 1.07 | - | - | - | - | - |
| Unified IO2 [4] | 11.64 | 34.28 | 2.15 | 2.40 | 1.02 | - | - | - | - | - |
| VideoSALMONN [30] | 11.90 | 32.32 | 2.07 | 2.39 | 0.91 | - | - | - | - | - |
| MAGNET +VideoSALMONN-ZS | 29.11 | 83.60 | 3.93 | 4.66 | 2.59 | 27.37 | 76.19 | 3.69 | 4.30 | 2.45 |
| MAGNET +Unified IO2-ZS | 28.78 | 81.79 | 3.85 | 4.52 | 2.54 | 27.95 | 76.1 | 3.69 | 4.35 | 2.45 |
| MAGNET +Qwen 2.5 Omni -ZS | 30.54 | 85.56 | 4.01 | 4.73 | 2.64 | 28.49 | 81.74 | 3.85 | 4.57 | 2.54 |
| MAGNET +VideoSALMONN-FT | 52.30 | 144.40 | 6.20 | 7.46 | 3.96 | 49.24 | 136.86 | 5.96 | 7.19 | 3.81 |
| MAGNET +Unified IO2-FT | 53.66 | 146.38 | 6.28 | 7.58 | 4.00 | 51.45 | 142.56 | 6.12 | 7.34 | 3.91 |
| **MAGNET +Qwen 2.5 Omni-FT** | **55.82** | **153.98** | **6.53** | **7.84** | **4.15** | **53.69** | **146.30** | **6.28** | **7.56** | **4.01** |
| MAGNET +Gemini 1.5 Pro | 57.67 | 157.72 | 6.69 | 8.03 | 4.25 | 55.80 | 153.95 | 6.53 | 7.80 | 4.15 |

Table 2: **Response Alignment Scores.** Our proposed MAGNET offers significant gains over baseline approaches (first section) and our adapted baselines (second section) across multiple objective and subjective metrics on two dataset splits. B@4: BLEU@4, Cr: CIDEr, H Eval: Human Evaluation. Closed source model: as a reference for upperbound.

| | AVHaystacks-50 | | | | | | AVHaystacks-Full | | | | | |
|---|---|---|---|---|---|---|---|---|---|---|---|---|
| Method | MTGS_avg ↑ | SM ↓ | SH ↓ | SO ↓ | SFP ↓ | SFN ↓ | MTGS_avg ↑ | SM ↓ | SH ↓ | SO ↓ | SFP ↓ | SFN ↓ |
| MAGNET +VideoSALMONN-ZS | 0.48 | 0.35 | 0.34 | 0.35 | 0.31 | 0.25 | 0.45 | 0.41 | 0.33 | 0.43 | 0.36 | 0.33 |
| MAGNET +Unified IO2-ZS | 0.51 | 0.39 | 0.31 | 0.31 | 0.32 | 0.22 | 0.42 | 0.49 | 0.39 | 0.37 | 0.37 | 0.29 |
| MAGNET +Qwen 2.5 Omni -ZS | 0.54 | 0.37 | 0.28 | 0.32 | 0.28 | 0.21 | 0.49 | 0.43 | 0.34 | 0.39 | 0.33 | 0.27 |
| MAGNET +VideoSALMONN-FT | 0.81 | 0.12 | 0.16 | 0.19 | 0.18 | 0.11 | 0.75 | 0.13 | 0.18 | 0.23 | 0.19 | 0.14 |
| MAGNET +Unified IO2-FT | 0.79 | 0.14 | 0.16 | 0.17 | 0.18 | 0.14 | 0.72 | 0.15 | 0.18 | 0.20 | 0.21 | 0.18 |
| **MAGNET +Qwen 2.5 Omni-FT** | **0.83** | **0.11** | **0.13** | **0.14** | **0.15** | **0.09** | **0.79** | **0.13** | **0.16** | **0.19** | **0.19** | **0.12** |
| MAGNET +Gemini 1.5 Pro | 0.85 | 0.09 | 0.12 | 0.14 | 0.10 | 0.07 | 0.81 | 0.12 | 0.14 | 0.17 | 0.12 | 0.09 |

Table 3: **Grounding evaluation and Step-wise error results** on AVHaystack-50 and AVHaystack-Full datasets using MTGS and STEM (SM, SH, SO, SFP, SFN) metrics respectively.

| | AVHaystacks-50 | | AVHaystacks-Full | |
|---|---|---|---|---|
| Method | R@3 ↑ | R@5 ↑ | R@3 ↑ | R@5 ↑ |
| ImageBind-RAG [] | 78.22 | 82.57 | 60.41 | 66.13 |
| Text-RAG | 82.84 | 84.87 | 66.54 | 71.60 |
| Video-RAG | 85.26 | 88.52 | 69.83 | 73.52 |
| VideoRAG | 85.57 | 89.79 | 70.43 | 74.96 |
| **Ours** | **90.68** | **93.17** | **73.15** | **79.20** |

Table 4: **Retrieval Evaluation Scores** on AVHaystack-50 and AVHaystack-Full datasets.

## 4.3 Main Results

**Audio Visual QA.** Experimental results in Tab. 2 demonstrate that among FT models MAGNET +Qwen 2.5 Omni-FT, achieves best performance across all automatic and human evaluation metrics. On both AVHaystacks splits, it achieves the highest scores outperforming both ZS and fine-tuned variants of MAGNET combined with other AVLLMs (e.g., VideoSALMONN, Unified IO2).

**Grounding Evaluation and Step-wise Error Assessment.** Tab. 3 indicates that our approach substantially improves the grounding capabilities and reduces step-wise error rates of the open-source AVLLMs, as reflected from the MTGS_Avg and STEM values, respectively. To robustly validate our proposed STEM, we conduct a human evaluation and observe a strong correlation between human judgments and the metric. Refer to supplementary material for more details.

Notably, in both Tabs. 2 - 3, to provide an upper bound, we incorporate a strong closed-source model with powerful generative capabilities such as Gemini-1.5 -Pro [32] within multi-agent framework MAGNET, coupled with Salient Frame Selector module and report its performance. We observe that our best model MAGNET +Qwen 2.5 Omni-FT almost reaches the upper bound values for .

**Audio Visual Retrieval.** Our method sets a new benchmark on AV retrieval task across both the dataset splits (Tab. 4). Compared to existing approaches, it achieves the largest margin of improvement in R@3 and R@5, particularly on the more challenging AVHaystacks-Full, where gains over strong baselines like VideoRAG and Text-RAG are 2.7 points in R@3 and 7.6 points in R@5. These improvements indicate that our approach retrieves more relevant samples consistently and scales effectively to larger retrieval spaces. The results also highlight the benefit of leveraging richer modality integration and adaptive reasoning in our model over static retrieval pipelines.

The above results (Tabs. 2 - 4) demonstrate that employing our multi-modal RAG pipeline not only improves retrieval quality but also aligns better with human judgments.

## 4.4 Ablations

**Importance of modalities.** The results in Tab. 5 show that performance is generally best when both audio and visual modalities are used, highlighting the benefit of multi-modal information. Gemini-1.5-Pro consistently outperforms Qwen-2.5-Omni across all retrieval and response alignment scores metrics, indicating the benefits of MAGNET in formulating coherent and information-rich responses. Qualitative assessment indicates a strong correlation across tasks, underlining the utility of our RAG pipeline.

**Sampling strategy.** For both backbones, using SFS significantly improves performance across all metrics. For Qwen2.5-Omni-FT, switching from Uniform to SFS increases BLEU@4 score by 0.17

| Method | Audio | Visual | BLEU@4 ↑ | Text Sim ↑ | GPT Eval ↑ | Human Eval ↑ |
|---|---|---|---|---|---|---|
| MAGNET +Qwen-2.5-Omni-FT | ✗ | ✓ | 45.28 | 5.15 | 6.16 | 3.32 |
|  | ✓ | ✗ | 38.96 | 4.82 | 5.78 | 3.13 |
|  | ✓ | ✓ | **53.64** | **6.28** | **7.52** | **4.01** |
| MAGNET +Gemini-1.5-Pro | ✗ | ✓ | 48.48 | 5.55 | 6.64 | 3.57 |
|  | ✓ | ✗ | 42.94 | 5.15 | 6.14 | 3.32 |
|  | ✓ | ✓ | **55.80** | **6.53** | **7.85** | **4.15** |

Table 5: **Performance of MAGNET under different modality settings on AVHaystacks-Full.**

| Method | Uniform | SFS | BLEU@4 ↑ | Text Sim ↑ | GPT Eval ↑ | Human Eval ↑ |
|---|---|---|---|---|---|---|
| MAGNET +Unified-IO2-FT | ✓ | ✗ | 34.89 | 4.41 | 5.18 | 2.88 |
|  | ✗ | ✓ | 51.45 | 6.12 | 7.34 | 3.91 |
| MAGNET +Qwen-2.5-Omni-FT | ✓ | ✗ | 36.58 | 4.58 | 5.42 | 2.98 |
|  | ✗ | ✓ | 53.61 | 6.28 | 7.53 | 4.01 |
| MAGNET +Gemini-1.5-Pro | ✓ | ✗ | 41.94 | 4.74 | 5.64 | 3.08 |
|  | ✗ | ✓ | 55.87 | 6.53 | 7.81 | 4.15 |

Table 6: **Effect of sampling strategy.** We systematically analyse our design choice replacing the sampling algorithm with 3 models on AVHaystacks-Full.

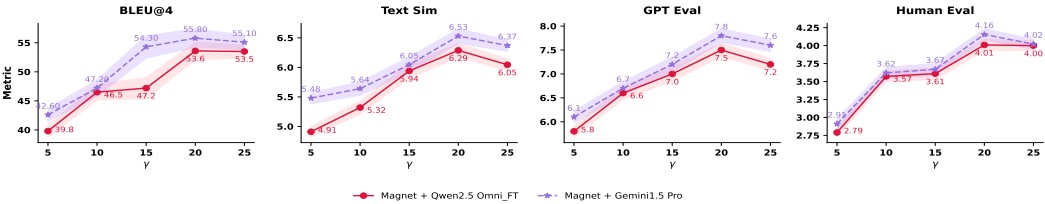

Figure 3: Effect of $\gamma$ on eval metrics for MAGNET $_{+Qwen\text{-}2.5\text{-}Omni\text{-}FT}$ and MAGNET $_{+Gemini\text{-}1.5\text{-}Pro}$.

and Human Eval score by 1.03. A similar trend is observed with Gemini 1.5 Pro where the best results are obtained with our proposed sampling strategy as seen in Tab. 6. These results underscore the advantage of semantically guided sampling over uniform strategies, as SFS more effectively captures informative segments, leading to better grounding, coherence, and human preference.

**Penalty hyperparameter.** Fig. 3 demonstrates a steady rise in performance across all metrics as $\gamma$ increases, although a slight dip in performance is observed at $\gamma = 25$, notably in BLEU@4 and Human Eval for MAGNET + Qwen-2.5-Omni-FT, potentially indicate the onset of overfitting or increased parameter sensitivity in that region. The varying magnitudes of the dip across metrics indicate that the effect of $\gamma$ is not uniform across different aspects of model performance.

### 4.5 Qualitative Results

Fig.4 showcases our system's ability to retrieve the most relevant videos and accurately localize temporal segments needed to answer a query using audio-visual cues. The meta-agent effectively highlights key instructional moments e.g., forming the fulcrum (Video 1), handshake shaping (Video 8), and diagonal pivoting (Video 1) in alignment with expert references. It also merges complementary segments from Videos 8 and 18 to describe finger placement, demonstrating robustness to redundancy and variation. Overall, MAGNET excels at retrieving, grounding, and synthesising evidence across videos into coherent, temporally aligned responses compared to a recent baseline – which fails to retrieve suitable videos and subsequently fails to temporally ground the salient regions.

**Additional qualitative and quantitative results are provided in the supplementary material**.

## 5 Related Works

**Video QA Benchmarks.** Video Question Answering (VidQA) involves answering natural language queries using visual content alone or in combination with other modalities like audio [33–36]. Early benchmarks such as MovieQA [37] relied heavily on subtitles, with minimal visual grounding [38]. Datasets like ActivityNet-QA [39] and How2QA [40] target visual understanding in daily and instructional contexts. More recent efforts, including NeXT-QA [41], Perception Test [42], STAR [43], and AGQA [44], emphasize spatio-temporal and causal reasoning. EgoSchema [45] extends VidQA to long-form egocentric videos using LLM-generated questions. Other benchmarks address longer video reasoning [46, 36] and specialized domains like instructional [47, 48] and egocentric content [49–52]. Despite progress, most VidQA datasets are constrained by limited modalities or fixed time windows. Our work enables large-scale, cross-video retrieval and multimodal reasoning, bridging VidQA and broader audio-visual understanding.

**MLLMs for Video Understanding.** Open-source LLMs [53–55] have enabled Video MLLMs that connect visual encoders to LLMs via projection bridges [56, 34, 57, 58]. While effective on short clips, they struggle with long videos due to context limits and temporal complexity [11]. To address this, long-context LLMs [59–62] and token compression [63, 15, 64] scale input capacity and support agent-based decomposition and retrieval [65–67]. MovieChat [15], for example, uses hierarchical memory for frame-level summarization. However, these models lack the ability to reason over audio-visual content across multiple videos and generate coherent responses. Our multi-agent framework addresses this gap.

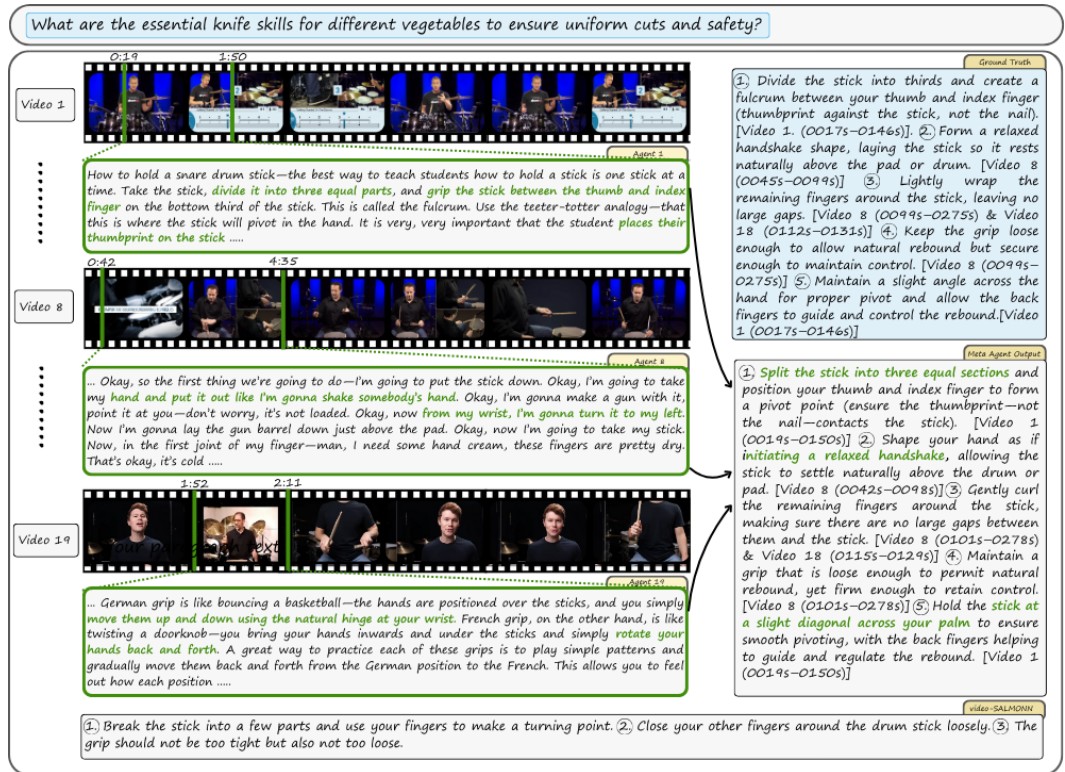

Figure 4: **Qualitative results.** Powered by efficient video retrieval pipeline and multi-agent configurations of MAGNET +Qwen-2.5-Omni-FT demonstrates strong reasoning abilities by first identifying the key videos followed by audio-visual temporal grounding to localise the salient regions across multiple videos when subjected to a how-to question.

**Retrieval Augmented Generation (RAG).** RAG improves generative models by integrating retrieval to inject external knowledge [68–73]. While well-studied in text domains [74–79], recent efforts have extended RAG to vision-language tasks [80–85]. MuRAG [81] uses non-parametric multimodal memory, and MIRAGE [82] employs CLIP-based retrieval. Other methods convert images to text via OCR, captioning, and detection before dense retrieval [86, 79]. Multimodal RAG has also been applied in domains like healthcare [87, 88], leveraging images as contextual input. While prior models focus on text or vision alone, MAGNET integrates off-the-shelf models with a custom retrieval-fusion pipeline for scalable audio-visual-language retrieval and generation.

## 6   Conclusions and Future Work

We introduced a novel benchmark and framework for evaluating LMMs in audio-visual retrieval and reasoning an area less explored than image or single-video based settings. Our task targets the challenging problem of retrieving relevant audio-visual segments from large video corpora, requiring joint temporal, auditory, and visual reasoning, akin to real-world multimedia search. To tackle this, we proposed MAGNET, a scalable retrieval-augmented generation system that identifies key moments across multiple videos and synthesizes grounded responses. It combines a sampling strategy, off-the-shelf models, and a multi-agent relevance scoring mechanism to extract and fuse salient content. Experiments show substantial gains over baselines, underscoring the promise of retrieval-augmented methods in audio-visual reasoning. We hope this work inspires richer benchmarks that push LMMs toward dynamic, temporally grounded multimodal understanding.

While MAGNET and AVHaystacks advance AV multi-video reasoning, several future directions remain. Replacing off-the-shelf components with end-to-end trainable modules could improve retrieval and frame selection. Enhancing agentic reasoning with collaborative mechanisms (e.g., planning or voting) may boost interpretability and performance. Lastly, integrating personalisation (e.g., user-driven retrieval) would support real-world applications like education and assistive tools.

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

# 🧲 MAGNET: A Multi-agent Framework for Finding Audio-Visual Needles by Reasoning over Multi-Video Haystacks

## Supplementary Material

**The supplementary is organised as follows**:

## A    Supplementary Video

In the supplementary video, we elaborate on our proposed task AVHaystacksQA with illustrative examples and demonstrate the intricacies involved in a multi-video linked QA setting. The video shows how the given question involves referring to multiple videos to obtain a comprehensive answer, followed by the meta agent summarising the responses to come up with the final answer. We also highlight the salient components of MAGNET and discuss the end-to-end flow. The use of headphones is recommended for a better audio-visual QA experience.

## B    Dataset Statistics

In this section, we provide additional details about AVHaystacks. Tab. B summarizes the topics from which the samples are collected, along with the number of questions per category and the corresponding video-to-question ratio. The benchmark comprises **103 hours** of video content from **500 video samples** across **27 diverse** categories, accompanied by carefully annotated QA pairs that temporally ground salient segments within the videos. **To the best of our knowledge, this is the first benchmark of its kind, as no prior work provides multi-video linked audio-visual QA pairs.**

Fig. 5 and 6 depict the distribution of total video hours across the various categories included in our benchmark. As shown, the samples are well distributed and sourced from a wide range of scenarios. A significant portion is drawn from *Travel Destinations*, *Education and Language Learning*, *Music and Performing Arts*, and *DIY and Creative Hobbies*—domains that demand strong audio-visual comprehension for effective temporally grounded AV QA.

Fig. 7 illustrates the distribution of sample durations (in minutes) within AVHaystacks. Most video samples fall within the 6–15 minute range, posing substantial challenges for temporal grounding tasks. Evaluation models must handle long context windows and effectively manage the increased complexity of processing extended multimodal sequences (vision and audio).

Additionally, Fig. 8 shows the distribution of the number of videos associated with each question. As observed, the majority of questions require referencing two or more videos to determine the answer, further emphasizing the complexity and richness of AVHaystacks.

Finally, it is to be noted that: (i) Our dataset annotation process requires manual inspection to ensure complete correctness. Although the sample curation process is automated, to ensure strict sanity we manually validate each sample, which is both tedious and time-consuming. (ii) The models evaluated in our study are already audio-visually informed, and we fine-tune them efficiently using LoRA adapters. Our salient frame selection module enhances context by focusing on key frames. With LoRA, we require fewer samples to fine-tune our audio-visual agents, as they are heavily pre-trained on audio-visual data. Importantly, we are only fine-tuning

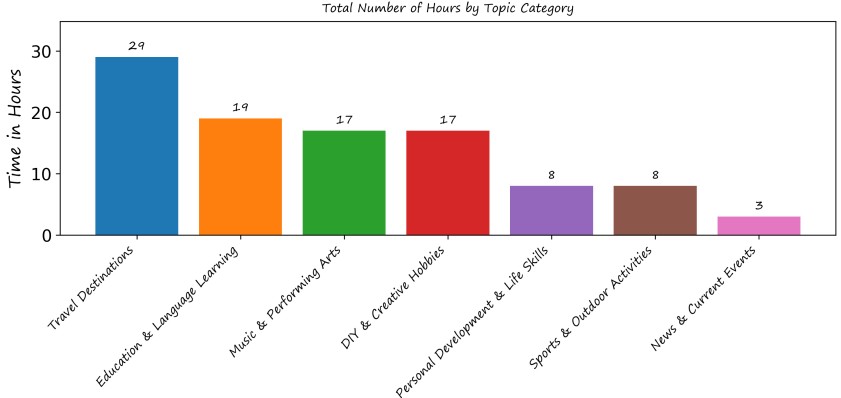

Figure 5: Number of hours of videos per topic category in AVHaystacks benchmark.

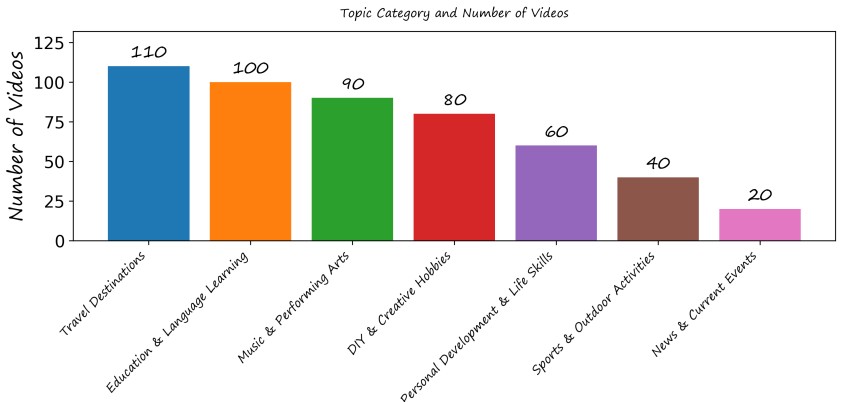

Figure 6: Number of videos per topic category in AVHaystacks benchmark.

the agents, not training the entire retrieval pipeline. Notably, the SFS module can be used with a closed-source model where finetuning was not done. As seen in Tables 2 and 3, these methods demonstrate strong performance even without any fine-tuning. Similar observations were made in recent literature [89–91] demonstrating that employing LoRA enables these models to perform well even with limited samples.

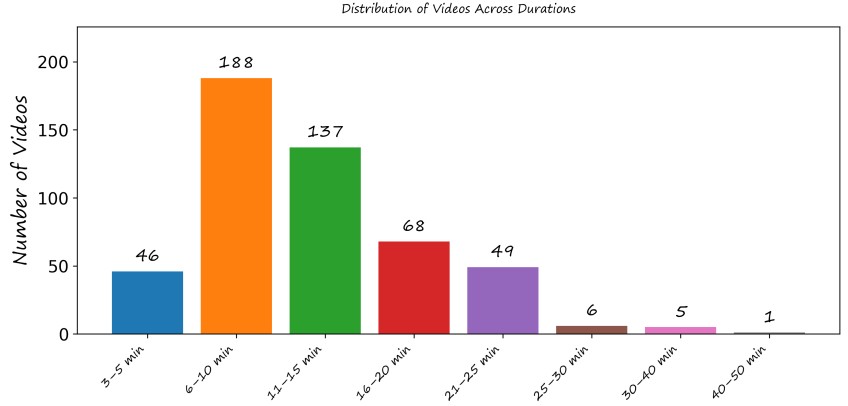

Figure 7: Distribution of videos based on their duration AVHaystacks benchmark.

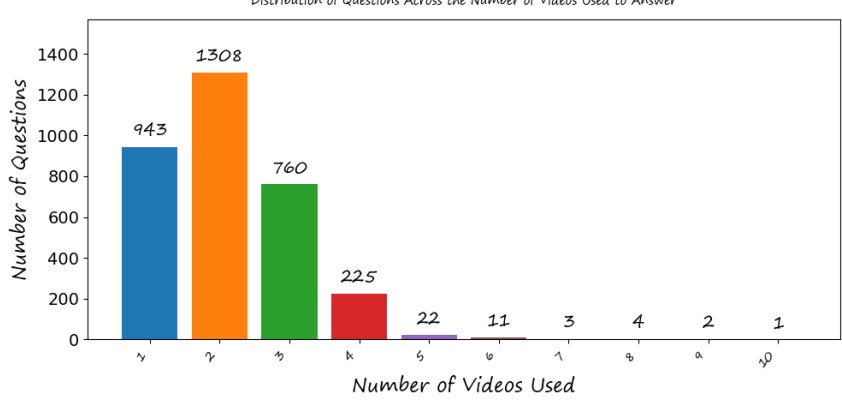

Figure 8: Number of videos referred to in each question AVHaystacks benchmark.

| Topic | # Videos | # Questions | Ratio Videos:Questions |
|---|---|---|---|
| General Cooking Tutorials | 20 | 117 | 1/6 |
| Learn Vocals | 20 | 51 | 2/5 |
| Learn English | 20 | 127 | 1/6 |
| Learn Arabic | 20 | 174 | 1/9 |
| Learn Chinese | 20 | 115 | 1/6 |
| Learn Urdu | 20 | 247 | 1/9 |
| Travel Turkey | 10 | 78 | 1/8 |
| Travel Brazil | 20 | 78 | 1/4 |
| Travel UAE | 20 | 105 | 1/5 |
| Travel Hawaii | 20 | 105 | 1/5 |
| Travel USA | 20 | 106 | 1/5 |
| Travel Italy | 20 | 106 | 1/5 |
| Sing Beatbox | 20 | 129 | 1/6 |
| Learn Opera | 20 | 123 | 1/6 |
| DIY | 20 | 187 | 1/9 |
| Learn Sketching | 20 | 89 | 1/4 |
| 3D Printing | 20 | 189 | 1/9 |
| Public Speaking | 20 | 97 | 1/5 |
| First-Aid | 20 | 125 | 1/6 |
| Self Defense | 20 | 126 | 1/6 |
| Soccer Analysis | 20 | 139 | 1/7 |
| News - Tornado | 20 | 112 | 1/5 |
| Sign Language | 20 | 107 | 1/5 |
| Hiking/Backpacking | 20 | 151 | 1/7 |
| Playing Music - Drums | 20 | 80 | 1/4 |
| Playing Music - String Instrument | 5 | 34 | 1/7 |
| Playing Music - Piano | 5 | 50 | 1/10 |
| **Total** | **500** | **3147** | **1/6** |

Table 7: Distribution of videos and QA pairs across different topics.

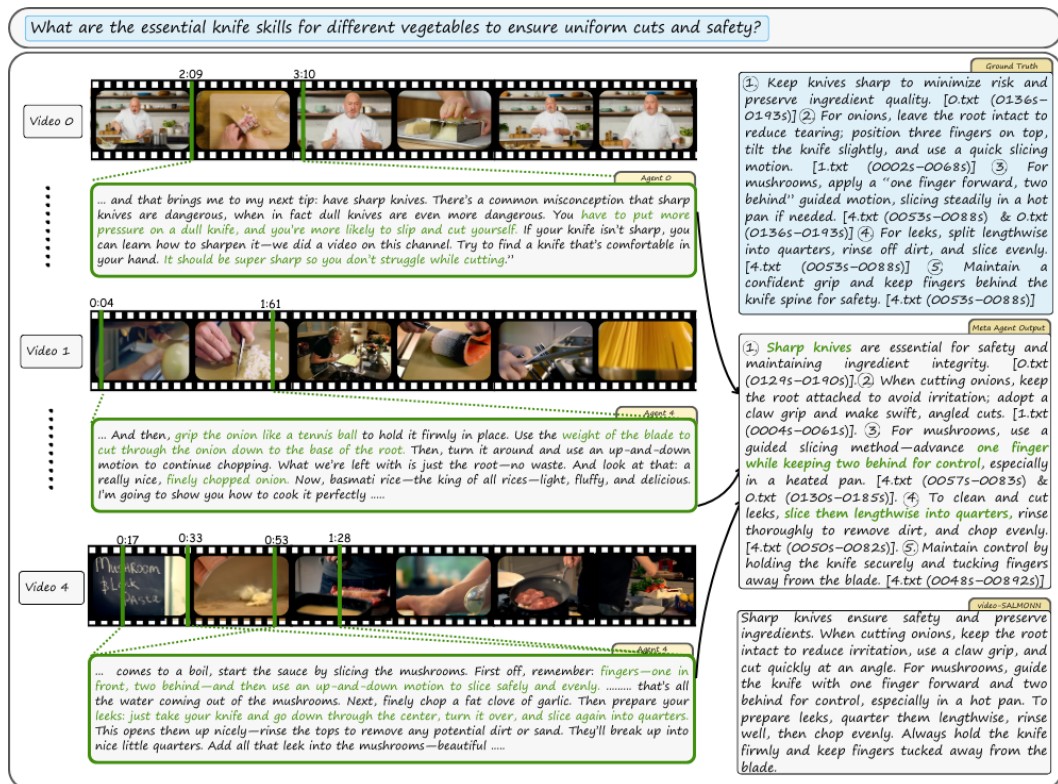

Figure 9: Performance comparison of MAGNET and Video-SALMONN on cooking video tutorial.

## C Qualitative Results

The qualitative examples in Fig. 9 to Fig. 13 showcase a variety of scenarios, including cooking demonstrations, language-speaking tutorials, and 3D printing lessons. In these examples, we compare the responses generated by baseline models with those produced by MAGNET. As illustrated, AVLLMs enhanced with our retrieval and multi-agent audio-visual reasoning modules consistently outperform the baselines on AVHaystacksQA. While the baseline models often struggle to identify and retrieve the most informative video segments, our method accurately selects the most relevant videos for each question, exhibiting strong audio-visual comprehension and reasoning capabilities. Furthermore, the temporal windows identified by our approach are generally precise and focused, reflecting a nuanced cross-modal understanding and effective temporal grounding across videos.

## D More Ablation Results

**Top-k Video Selection.** The top-$k$ selection experiment on AVHaystacks-Full shows (Tab. 8) that both Qwen-2.5-Omni-FT and Gemini-1.5-Pro achieve peak performance at $k = 6$, where BLEU4, Text Similarity, GPT Eval, and STEM-Missing metrics all indicate optimal retrieval and generation quality. Gemini-1.5-Pro consistently outperforms Qwen across all $k$ values, demonstrating stronger contextual understanding and output coherence. While increasing $k$ generally improves performance by reducing missed content, values beyond $k = 6$ offer slightly diminishing returns and may introduce noise. Further, it leads to an increase in compute (since more AVLLM agents are required). These findings highlight the importance of selecting an appropriate $k$ and leveraging more capable language models like Gemini for effective multimodal retrieval.

**Meta Agents.** Tab. 9 analyses the impact of different meta agents within the MAGNET framework on the AVHaystacks-Full benchmark. Across both base models: Qwen-2.5-Omni-FT and Gemini-1.5-Pro performance consistently improves as stronger meta agents are used, with Gemini achieving the best results in all metrics (BLEU4, Text Sim, GPT Eval, and STEM-Order). Notably, Gemini, as both the meta agent and core model, yields the highest overall performance, highlighting its superior capability in segment selection, coherence, and multimodal reasoning. These results underscore the critical role of the meta agent in guiding high-quality content synthesis.

**Frame Sampling Function.** Tab. 10 compares the effect of different frame sampling functions (SFS) on the AVHaystacks-Full benchmark using two base models: MAGNET $_{+Qwen-2.5-Omni-FT}$ and MAGNET $_{+Gemini-1.5-Pro}$.

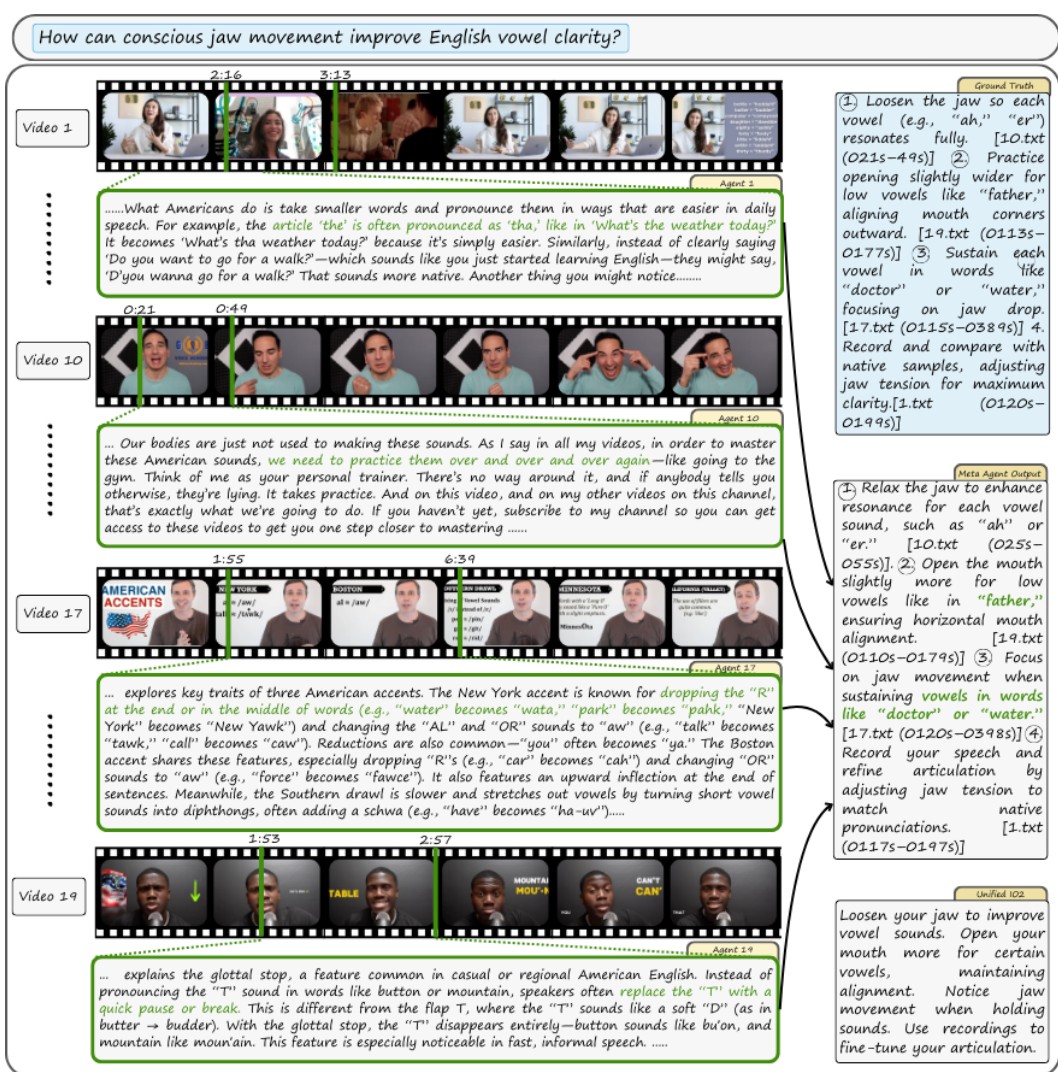

Figure 10: Performance comparison of MAGNET and Unified IO2 on english pronunciation tutorial.

Across both models, we observe that the proposed SFS function (highlighted), which dynamically scales dissimilarity using an inverse sine function, consistently outperforms cosine and exponential alternatives. For instance, with Qwen-2.5-Omni-FT, the proposed SFS achieves the highest scores across all metrics. Similarly, when paired with Gemini-1.5-Pro, the same function yields the best performance, achieving high BLEU4 and a notable MTGS$_{avg}$. These results suggest that the sinusoidal-inverse-based SFS is more effective at capturing temporal importance for reasoning tasks, likely due to its sharper penalisation of semantically redundant or temporally proximal frames. This indicates that thoughtful frame selection plays a critical role in enhancing downstream multimodal generation quality.

**Number of Video Frames Selection.** Tab.11 analyzes the impact of varying the number of uniformly sampled frames $m$ on AVHaystacks-Full using two base models. Across both models, performance consistently improves as the number of frames increases from 15 to 75. For instance, with Qwen-2.5-Omni-FT, BLEU@4 rises from 45.89 to 53.61, and MTGS$_{avg}$ improves substantially from 0.43 to 0.83, indicating that denser frame sampling leads to more accurate and temporally grounded responses. Similarly, Gemini-1.5-Pro shows a consistent upward trend, reaching its peak performance at $m = 75$. These results suggest that richer frame coverage provides stronger contextual grounding for both models, reinforcing the need for higher temporal granularity in multi-video audio-visual reasoning tasks.

**Text Similarity Threshold.**

Tab. 12 presents the effect of varying the text similarity threshold $\tau_s$ used in frame filtering on AVHaystacks-Full. For both methods, performance peaks at $\tau_s = 0.5$, with **BLEU@4**, **Text Sim**, and **GPT Eval** achieving the

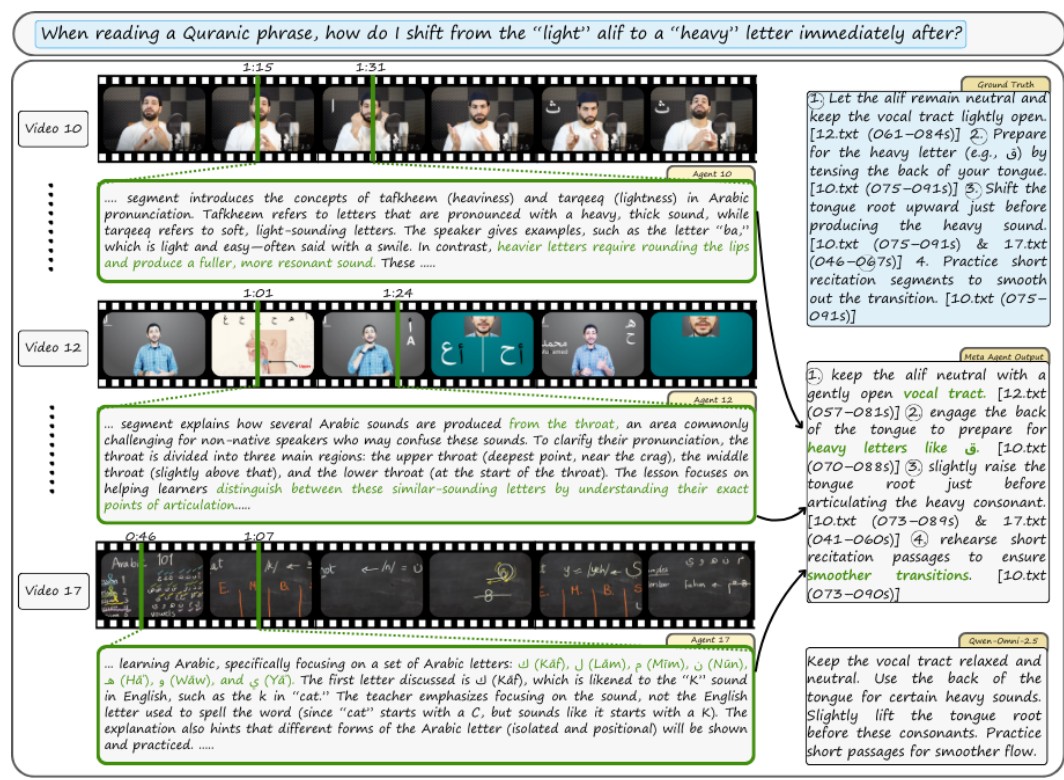

Figure 11: Performance comparison of MAGNET and Qwen-2.5-Omni on Urdu pronunciation tutorial.

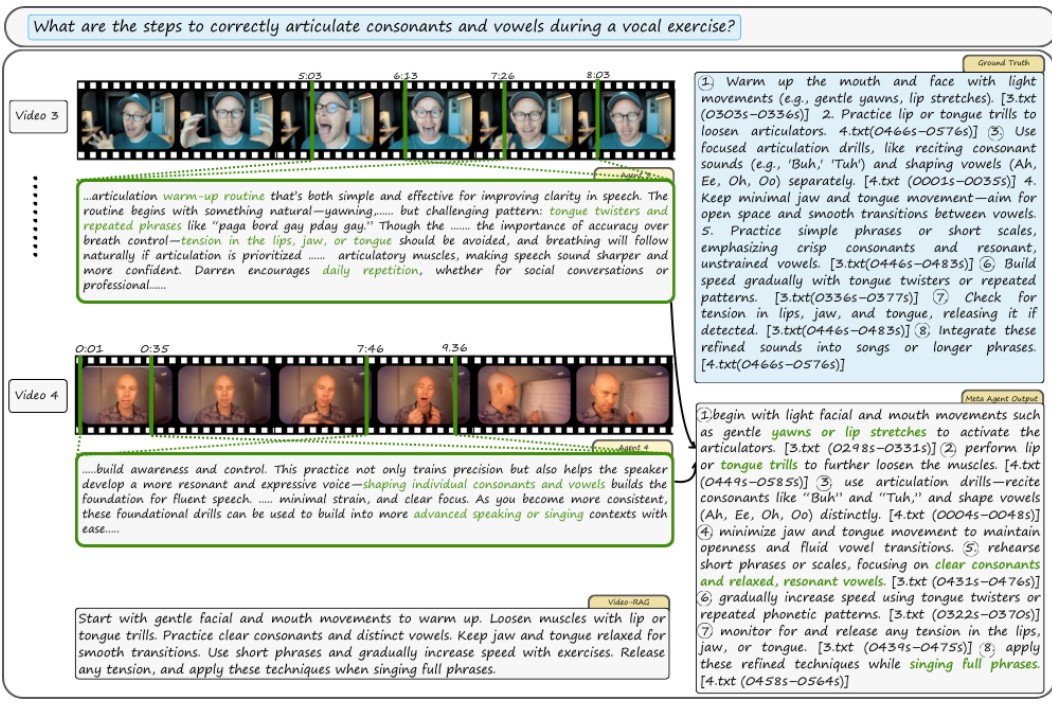

Figure 12: Performance comparison of MAGNET and Video-RAG on vocal exercise tutorial.

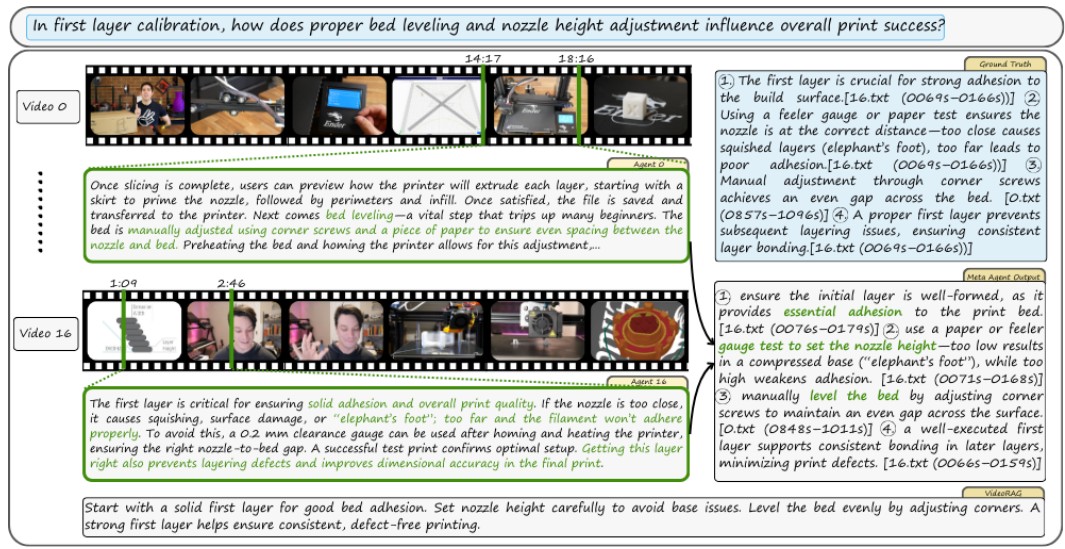

Figure 13: Performance comparison of MAGNET and VideoRAG on 3D printing tutorial.

highest scores, while the STEM-Missing and STEM-Order errors are minimized. Specifically, **Qwen-2.5-Omni-FT** sees improvements in BLEU@4 from **51.32** to **53.61** and a drop in STEM-Missing from **0.15** to **0.13** as $\tau_s$ increases from 0.3 to 0.5. Similarly, **Gemini-1.5-Pro** reaches optimal performance at $\tau_s = 0.5$, with the best overall textual coherence and minimal temporal grounding errors. However, setting $\tau_s = 0.7$ degrades performance across metrics, suggesting that overly strict filtering removes useful context. These findings highlight the importance of balancing informativeness and precision in frame selection by carefully tuning the similarity threshold.

## E   Human Evaluation on STEM:

Tab. 13 presents a comparative evaluation of various MAGNET model configurations using both automated STEM and human evaluation metrics averaged across 20 raters (Cohen's $\kappa = 0.82$). In particular, the fine-tuned MAGNET $_{+ Qwen 2.5 Omni-FT}$ model achieves strong overall performance, outperforming other models in most categories. Specifically, it ties for the lowest SM, achieves the lowest SH, and second-lowest SO scores under STEM, indicating improved semantic and syntactic alignment. It also performs competitively in human evaluations, with HM, HH, and HO scores close to or better than all other non-proprietary models. Although MAGNET $_{+ Gemini 1.5 Pro}$ demonstrates slightly superior performance across several metrics. These results underscore the efficacy of fine-tuning with Qwen 2.5 Omni, particularly in aligning model outputs with human judgments.

| Method | $K$ | BLEU@4 ↑ | Text Sim ↑ | GPT Eval ↑ | STEM-Missing ↓ |
|---|---|---|---|---|---|
| MAGNET $_{+Qwen-2.5-Omni-FT}$ | 1 | 46.67 | 4.67 | 5.37 | 0.29 |
| | 3 | 50.44 | 5.93 | 6.88 | 0.20 |
| | **6** | **53.61** | **6.28** | **7.53** | **0.13** |
| | 10 | 53.38 | 6.18 | 7.36 | 0.14 |
| MAGNET $_{+Gemini-1.5-Pro}$ | 1 | 49.80 | 4.84 | 5.04 | 0.27 |
| | 3 | 52.09 | 5.22 | 6.31 | 0.19 |
| | **6** | **55.87** | **6.53** | **7.81** | **0.12** |
| | 10 | 55.63 | 6.41 | 7.64 | 0.12 |

Table 8: **Top-k selection** on AVHaystacks-Full.

## F   More Details on Benchmark Construction

In this section, we provide further details on AVHaystacks construction. We outline the steps involved in data preparation in Fig. 14. The complete benchmark creation pipeline details.

| Method | Meta Agent | BLEU@4 ↑ | Text Sim ↑ | GPT Eval ↑ | STEM-Order ↓ |
|---|---|---|---|---|---|
| MAGNET +Qwen-2.5-Omni-FT | Reka | 51.38 | 5.91 | 6.88 | 0.25 |
| | Qwen-2.5-Omni-FT | 51.98 | 5.98 | 6.96 | 0.24 |
| | Claude | 52.29 | 6.05 | 7.34 | 0.22 |
| | GPT | 52.93 | 6.11 | 7.37 | 0.20 |
| | **Gemini** | **53.61** | **6.28** | **7.53** | **0.19** |
| MAGNET +Gemini-1.5-Pro | Reka | 54.89 | 6.02 | 6.80 | 0.22 |
| | Claude | 55.02 | 6.13 | 7.09 | 0.20 |
| | GPT | 55.32 | 6.20 | 7.48 | 0.19 |
| | **Gemini** | **55.87** | **6.53** | **7.81** | **0.17** |

Table 9: **Effect of meta agents** on AVHaystacks-Full.

| Method | SFS Function | BLEU@4 ↑ | Text Sim ↑ | GPT Eval ↑ | MTGS$_{avg}$ ↑ |
|---|---|---|---|---|---|
| MAGNET +Qwen-2.5-Omni-FT | $\Delta_{ab} = \gamma \left( \cos \left( \frac{\pi}{2} |a-b| \right) - 1 \right); \; \gamma = 10$ | 52.85 | 5.93 | 7.24 | 0.79 |
| | $\Delta_{ab} = \gamma \left( e^{\lambda |a-b|} - 1 \right); \; \gamma = 10, \lambda = 5$ | 53.18 | 6.03 | 7.47 | 0.81 |
| | $\Delta_{ab} = \gamma \left( \frac{1}{\sin\left( \frac{\pi}{2} |a-b| \right)+1} - 1 \right); \; \gamma = 20$ | **53.61** | **6.28** | **7.53** | **0.83** |
| MAGNET +Gemini-1.5-Pro | $\Delta_{ab} = \gamma \left( \cos \left( \frac{\pi}{2} |a-b| \right) - 1 \right); \; \gamma = 10$ | 55.20 | 6.01 | 7.33 | 0.80 |
| | $\Delta_{ab} = \gamma \left( e^{\lambda |a-b|} - 1 \right); \; \gamma = 10, \lambda = 5$ | 55.79 | 6.15 | 7.62 | 0.82 |
| | $\Delta_{ab} = \gamma \left( \frac{1}{\sin\left( \frac{\pi}{2} |a-b| \right)+1} - 1 \right); \; \gamma = 20$ | **55.87** | **6.53** | **7.81** | **0.85** |

Table 10: **Effect of different frame sampling functions** on AVHaystacks-Full.

| Method | $m$ | BLEU@4 ↑ | Text Sim ↑ | GPT Eval ↑ | MTGS$_{avg}$ ↑ |
|---|---|---|---|---|---|
| MAGNET +Qwen-2.5-Omni-FT | 15 | 45.89 | 4.46 | 5.10 | 0.43 |
| | 50 | 49.27 | 5.19 | 6.03 | 0.62 |
| | **75** | **53.61** | **6.28** | **7.53** | **0.83** |
| MAGNET +Gemini-1.5-Pro | 15 | 47.36 | 4.92 | 5.44 | 0.51 |
| | 50 | 51.56 | 5.81 | 6.62 | 0.69 |
| | **75** | **55.87** | **6.53** | **7.81** | **0.85** |

Table 11: **Effect of number of frames selection** on AVHaystacks-Full.

| Method | $\tau_s$ | BLEU@4 ↑ | Text Sim. ↑ | GPT Eval ↑ | STEM-Missing ↓ | STEM-Order ↓ |
|---|---|---|---|---|---|---|
| MAGNET +Qwen-2.5-Omni-FT | 0.3 | 51.32 | 5.75 | 6.68 | 0.15 | 0.24 |
| | **0.5** | **53.61** | **6.28** | **7.53** | **0.13** | **0.19** |
| | 0.7 | 50.67 | 5.52 | 6.43 | 0.19 | 0.22 |
| MAGNET +Gemini-1.5-Pro | 0.3 | 54.13 | 6.31 | 7.56 | 0.13 | 0.22 |
| | **0.5** | **55.87** | **6.53** | **7.81** | **0.12** | **0.17** |
| | 0.7 | 53.95 | 6.30 | 7.51 | 0.17 | 0.20 |

Table 12: **Text similarity threshold** on AVHaystacks-Full.

## F.1 Dataset Examples

Please find example videos with QAs from different categories as shared in the supplementary zip (refer to the *'AVHaystacks-dataset-samples'* folder). **The video files are compressed to fit within the supplementary material size limit**. Samples are collected from different areas (how-to, musical lessons, news etc) making our benchmark extremely diverse and considerably challenging for the models. The purpose of curating samples from such diverse sources is to robustly evaluate every model on their generalization capabilities.

| Method | StEM | | | Human Eval. | | |
|---|---|---|---|---|---|---|
| | SM ↓ | SH ↓ | SO ↓ | HM ↓ | HH ↓ | HO ↓ |
| MAGNET +VideoSALMONN-ZS | 0.41 | 0.33 | 0.43 | 0.39 | 0.37 | 0.41 |
| MAGNET +Unified IO2-ZS | 0.49 | 0.39 | 0.37 | 0.46 | 0.35 | 0.36 |
| MAGNET +Qwen 2.5 Omni -ZS | 0.43 | 0.34 | 0.39 | 0.45 | 0.37 | 0.42 |
| MAGNET +VideoSALMONN-FT | 0.13 | 0.18 | 0.23 | 0.15 | 0.17 | 0.21 |
| MAGNET +Unified IO2-FT | 0.15 | 0.18 | 0.20 | 0.19 | 0.23 | 0.22 |
| **MAGNET +Qwen 2.5 Omni-FT** | **0.13** | **0.16** | **0.19** | **0.17** | **0.18** | **0.16** |
| MAGNET +Gemini 1.5 Pro | 0.12 | 0.14 | 0.17 | 0.14 | 0.18 | 0.13 |

Table 13: **StEMvs Human** on AVHaystack-Full dataset. SM: StEM- Missing, SH: StEM-Hallucination, SO: StEM- Order, HM: Human Eval - Missing, HH: Human Eval. - Hallucination, HO - Human Eval. - Order

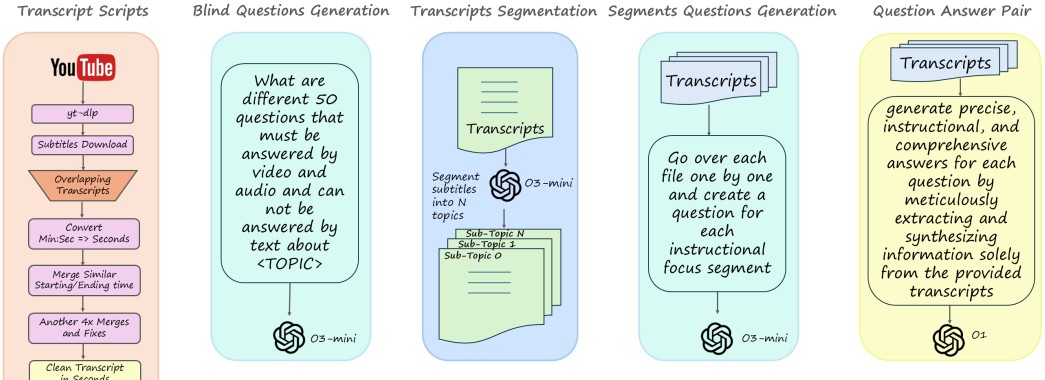

Figure 14: Steps involved in benchmark data collection.

## F.2 Pipeline overview

Our multi-step data collection strategy is as follows in sequence: (1) download and curate 500 videos that satisfy the four-modality filter (Dataset Selection); (2) issue 50 blind prompts per topic (Listing 1) to GPT *without* revealing transcripts to encourage cross-video reasoning; (3) repair caption timing (Alg. 1) and convert HH:MM:SS→seconds for uniform indexing; (4) segment transcripts into sub-topics (Listing 4) and create one segment-aware question each (Listing 2); and (5) assemble QA pairs whose answers cite at least two videos (Listing 5).

## F.3 Question-pipeline specifics

(i) *Blind phase*: prompt GPT with Listing 1; discard any question whose answer is general knowledge information or an answer that does not depend on audio and visuals; (ii) *Segment phase*: To expand the number of questions, we used segmented-phase questions. For each sub-topic, prompt GPT with Listing 2; ensure the generated question references audio, visual, and caption tokens.

Listing 1: Blind questions generation prompt

```
I am collecting data about how to <TOPIC>.
What are different 50 questions
that must be answer by a video, audio
and can not be answered by text only?
```

Listing 2: Segments-based questions generation

```
Go over each file one by one and create
a question for each segments
```

## F.4 Caption repair and segmentation

(1) *Overlap fix*: downloaded subtitles had mis-matched intervals which needs to be fixed (example in Listing 3); (2) *Normalization*: map all times to seconds and drop duplicate caption lines; (3) *Segmentation*: apply the template in Listing 4 and retain segments whose duration lies in $[15, 120]$ s.

Listing 3: Transcripts timing mismatches

```
00:00:00 --> 00:00:**03**
hi everyone welcome to my youtube channel
00:00:**01** --> 00:00:05
about parrots. i am david. i'm here with
```

Listing 4: Transcripts segmentation output format

```
Segment [Segment Number]
Time: [Start Seconds] --> [End Seconds]
Title: [Concise Title]
Details:
   - Instructional Focus: Brief description
   - Key Steps and details:
      - [Step / Detail 1]
      - [Step / Detail 2]
      - [Step / Detail 3]
      - [Step / Detail 4]
      - [Step / Detail 5]
   - Audio Cues: Audio elements description
```

## F.5 QA-pair generation

After transcripts are segmented, GPT is prompted to produce a triple consisting of (i) the question itself, (ii) a step-by-step answer, and (iii) a list of ⟨videoID, start, end⟩ references that support each step. Crucially, every answer must draw evidence from *multiple* videos, making cross-clip reasoning a core requirement of AVHaystacks (see Listing 5).

Listing 5: Question-Answer pair output format

```
Question 1?
Answer:
   1) step 1
   2) step 2
   3) step 3
   4) step 4
   5) step 5
References:
  1.txt 0017s > 0074s, 8.txt 0045s > 0270s,
  2.txt 0050s > 0100s, 3.txt 0110s > 0150s
```

# G   SFS Prompt

Below, we add the prompt used to select the key frames using the SFS algorithm. We provide step-by-step, clear instructions about the task, the reasoning process, and the expected output. Among various other prompts employed, this one produces the best results.

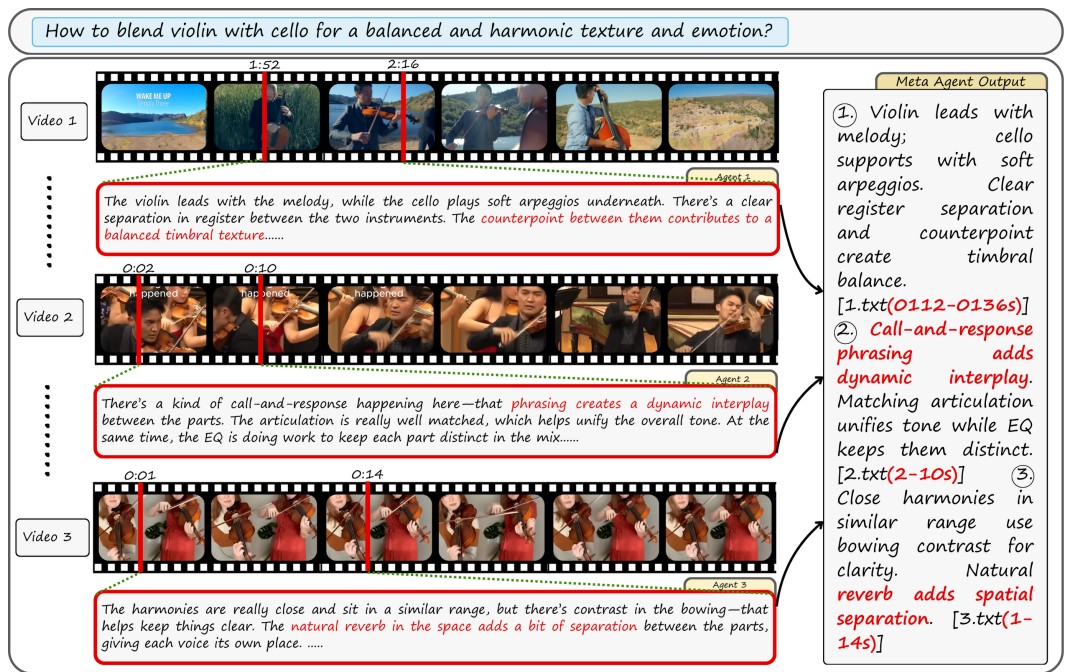

Figure 15: **Failure case of MAGNET.**

---

**Salient Frame Selection Prompt**

**Task:** Given a question and a set of key frames extracted from a video, identify the most relevant frames that best support answering the question.

**Step 1: Reasoning Process** Explain your selection by considering the following factors (in order of importance):
1) Presence of objects or actions explicitly mentioned in the question
2) Scenes that clearly align with the question's context
3) Visual elements directly related to the question details
4) Location or background context, even if the main object/action is not visible
5) Semantically related or typically co-occurring objects
6) Human motion or activity suggesting relevant events

**Step 2: Output** List the selected image indices using the format: [idx1, idx2, idx3, ...]

The objective is to select visual evidence useful for answering the question, not to answer the question itself.

## H  Failure case

Fig. 15 illustrates a failure case of MAGNET. Owing to the visual and auditory similarity between a violin and a cello, the retrieval module fails to accurately identify segments where both instruments are played simultaneously. As shown, the selected segment from the first video features only the cello, omitting the presence of the violin. The other two retrieved videos result from incorrect retrieval, leading to erroneous temporal grounding and highlighting a limitation of the system in distinguishing between acoustically and visually similar sources.

## I  Implementation Details

Training is done for 5 epochs on 4 A100 GPUs and the best checkpoint is selected for evaluation. Following the success of Low-Rank Adaptation (LoRA), we apply it with a rank of 8 and an alpha value of 32 for fine-tuning. AdamW is used as optimiser with a learning rate of 1e-4. We use a per-device batch size of 1 and a gradient accumulation step of 16. A cosine learning rate scheduler is employed with a warmup ratio of 0.05.

| Dataset | MS | TA | MVL | AVR | AVD | RQA | AVQA | LC |
|---|---|---|---|---|---|---|---|---|
| *Video Datasets* | | | | | | | | |
| Video-Bench [116] | ✓ | ✗ | ✗ | ✗ | ✗ | ✗ | ✗ | ✗ |
| EgoSchema [45] | ✓ | ✗ | ✗ | ✗ | ✗ | ✗ | ✗ | ✓ |
| MVBench [34] | ✓ | ✗ | ✗ | ✗ | ✗ | ✗ | ✗ | ✓ |
| MMBench-Video [117] | ✗ | ✗ | ✗ | ✗ | ✗ | ✗ | ✗ | ✓ |
| *Audio-Visual Datasets* | | | | | | | | |
| AVOdesseyBench [118] | ✓ | ✗ | ✗ | ✗ | ✗ | ✗ | ✓ | ✗ |
| OmniBench [119] | ✓ | ✗ | ✗ | ✗ | ✗ | ✗ | ✓ | ✗ |
| LongVALE [120] | ✓ | ✓ | ✗ | ✗ | ✓ | ✗ | ✓ | ✓ |
| AVHBench [121] | ✓ | ✗ | ✗ | ✗ | ✓ | ✗ | ✓ | ✗ |
| **AVHaystacks (Ours)** | ✓ | ✓ | ✓ | ✓ | ✓ | ✓ | ✓ | ✓ |

Table 14: **Comparison with prior video/audio-visual benchmarks.** MS: Model-Assisted; TA: Temporal Annotation; MVL: Multi-Video Linkage; AVR: Audio-Visual fine-grained Reasoning; AVD: Audio-Visual Description; RQA: Retrieval-based QA Answering; AVQA: Audio-Visual QA; LC: Long Context, where QA context spans over 5 mins.

# J   More related works

**Large Multimodal Models.** Large Multimodal Models (LMMs) have advanced significantly in understanding and reasoning over single and multiple images [92, 93, 24, 94, 95], expanding vision-language capabilities across diverse tasks and domains [96–99]. Their strength lies in large-scale cross-modal alignment and powerful language modeling. However, LMMs still struggle with scaling to large image or video collections [82, 100] due to computational and representational challenges. Retrieval-based approaches address this by enabling efficient access, processing, and reasoning over extensive multimedia content, including video and audio.

**(Audio)Video Benchmarks.** We compare AVHaystacks with recent audio-visual and video QA benchmarks in Tab. 14. As shown, most existing benchmarks do not offer multi-video linked QA annotations, making AVHaystacks the first of its kind and inherently more challenging. Our data collection framework introduces a scalable, semi-automated, and richly annotated pipeline in real-world settings, providing significant advantages for future research. We hope this benchmark will inspire the community to explore and advance work in this promising direction.

**Audio-Visual Learning.** Powered by improved models and high-quality annotated data, audio-visual learning has advanced significantly in areas such as cross-modal generation [101–104], representation learning [105–108], multimodal large language models [109, 2, 110, 30, 57, 6, 111], and cross-modal integration [112–114]. Recent works have contributed to cross-modal generation by leveraging visual and/or language context to generate coherent, complex audio [101, 102]. The work on active audio-visual separation and embodied agents highlights the role of motion and egocentric perception in learning robust representations. These ideas extend naturally to audio-visual LLMs [115, 2], where perceptually grounded models interact with dynamic environments.

# K   Human Study Details

We conducted a human study involving 20 participants to evaluate the following: (i) the correctness and reliability of the samples collected in AVHaystacks, (ii) the quality of responses generated by MAGNET, and (iii) the correlation between the proposed metric STEMand human evaluation. Application details are presented in Fig. 16, and user consent information is provided in Fig. 17.

Each participant received detailed instructions outlining the goals of the study and their specific tasks. They were shown several samples obtained through our semi-automated data collection strategy and asked to rate the quality of each sample on a scale from 1 to 5. The aggregated ratings indicate high relevance and correctness, with an average score of 4.6/5 for the collected samples. Participants also evaluated the responses generated by MAGNET on the proposed AVHaystacksQA, as discussed in the main paper.

The user study protocol was approved by the Institutional Review Board (IRB). No personal information was collected, stored, or shared at any stage of the study.

**Institutional Review Board**

**USER STUDY APPLICATION**

1. **Abstract:**

This study focuses on the evaluation of the quality of collected samples for a benchmark dataset, results generated by our novel framework for the project titled "MAGNET: A Multi-agent Framework for Finding Audio-Visual Needles by Reasoning over Multi-Video Haystacks" and assessing the quality of the proposed metric. We mainly use this technique for AI powered audio-visual processing applications. The evaluation is done by asking human participants to rate the quality of the samples and model responses.

2. **Subject Selection:**

   a. **Recruitment:** Hidden for anonymity

   b. **Eligibility Criteria:** Anyone with normal hearing and vision, is at least 18 years old, and is proficient in English is eligible. Participants with corrected-to-normal hearing and vision will also be eligible.

   We have two screening questions. Our screening question are 1) "Do you have normal hearing / corrected-to-normal hearing?" 2) "Are you proficient in English?". The participant who answers yes to both these questions will be allowed to continue with the survey.

   c. **Rationale:** Evaluations from people without normal hearing will bias our study results. Participants also need to be at least 18 years old and need to be proficient in English to understand the speech contents being played.

   d. **Enrollment Numbers:** 50 people max

   e. **Rationale for Enrollment Numbers:** This number gives reliable statistical results

3. **Procedures:**

   Using a headset is recommended possibly accompanied by videos and giving your preferences for each of them. The expected time to finish the procedure is 30 minutes. We are performing a survey/questionnaire and subject will only complete once. We will ask eligibility questions first and we will not collect survey data from ineligible participants. We will delete eligibility screening answers immediately for ineligible participants. You need to perform the following three tasks:

   First, the procedures involve you going over series of videos and assess the quality of the samples in the presented benchmark. Each sample has a question, a collection of videos and a response to it. Your task is to go over them carefully and rate the quality of the samples on a scale of 1-10 with 1 being the lowest. The samples are from diverse scenarios involving how-to, travelvlogs, new reading etc.

   Secondly, you need to go over model responses and rate them based on their accuracy, order, cohrence with the question asked.

---

   Finally, you need to evaluate the quality of our proposed metric and rate three aspects hallucination, order and missing steps.

4. **Risks:**

   There are no known risks.

5. **Benefits:**

   There are no direct benefits to participants. This study will provide researchers from audio-visual community a better understanding of how good the multi-video linked QA pipeline works and also evaluate them on various challenging cases.

6. **Confidentiality:**

   We do not collect information that can identify the participants. Any data collected will be stored on a password protected computer and will be securely wiped in 2 years from the day of creation. Only the investigators of this study have access to the data.

7. **Consent Process:**

   In our online study, we will first present screening questions before consenting to ensure that ineligible participants are not enrolled in your study. Then we present consent information to our participants, and they need to read and click a button that says "I agree" to indicate their consent and continue to our questions. We request a waiver of written consent for our online study based on following facts:
   (1) Our research only requires subjects to listen and view to videos, which involves no more than minimal risk to the subjects
   (2) We present participants with consent information electronically before the study. Subjects can choose to not continue if they do not give their consent. The waiver of written consent will not adversely affect the rights and welfare of the subjects.
   (3)      The research could not practically be carried out without the waiver or alteration because we need to collect responses from people in other regions in the .....
   and will not be able to collect signatures from each subject.
   (4)      We will provide our contact information during the study and encourage the subjects to contact us with any questions or concerns and they will be provided with additional pertinent information after participation.
   (5)      Participant can save their signed consent form. For a fair comparison, we will not use any deception.

8. **Conflict of Interest:**

   No conflict of interest.

9. **HIPAA Compliance:**

Not Applicable.

10. **Research Outside of the United States:**

Not Applicable.

11. **Research Involving Prisoners:**

Not Applicable.

12. **SUPPORTING DOCUMENTS**

Your Initial Application must include a **completed Initial Application Part 1 (On-Line Document),** the information required in items 1-11 above, and all relevant supporting documents including: consent forms, letters sent to recruit participants, questionnaires completed by participants, and any other material that will be presented, viewed or read to human subject participants.

The consent forms in your approved IRBNet PACKAGE must be used. When creating or editing your consent form, please provide the most recent IRBNet package number at the bottom, right corner of the consent form. This ensures you are using the most "uptodate" version of the form.

To find your IRBNet package number, go to the MY PROJECTS tab and click on the title of your project. In the PROJECT OVERVIEW page, your IRBNet package number will be listed at the top, next to your project title.

Figure 16: User study guidelines.

---

*Initials: _______ Date: _______*

**CONSENT TO PARTICIPATE**

| Project Title | A Multi-agent Framework for Finding Audio-Visual Needles by Reasoning over Multi-Video Haystacks |
|---|---|
| Purpose of the Study | This research is being conducted by ..... at ...... We are inviting you to participate in this research project because you have normal or corrected-to-normal hearing and visual abilities. The purpose of this research project is to evaluate the quality of our dataset, model response and correctness of our proposed metric. |
| Procedures | The procedures involve going over samples collected in the dataset and validate the correctness of the annotation. You will be presented with set of questions along with some videos and some responses relevant for them. You have to rate them based on their relevance, order and other aspects. You also need to go over the model response and evaluate its accuracy and provide a rating. Finally, you neede to rate the goodness of a metric by validating how good the score reported by the metric alignes with what you think is the correct response. The expected time to finish the procedure is 30 minutes. We will ask eligibility questions first and we will not collect survey data from ineligible participants. We will delete eligibility screening answers immediately for ineligible participants. |
| Potential Risks and Discomforts | There are no known risks from participating in this research study. |
| Potential Benefits | There are no direct benefits from participating in this research. We hope that, in the future, other people might benefit from this study through improved understanding of how to generate more realistic virtual sound for speech related applications. |
| Confidentiality | Any potential loss of confidentiality will be minimized by storing data in a password protected computer. No identifiable information will be collected and the researchers will be the only individuals to have access to the data. If we write a report or article about this research project, your identity will be protected. Your information may be shared with representatives of the ... or ... if you or someone else is in danger or if we are required to do so by law. |

Page 1 of 3    IRBNet Package: **1986481-1**    Initials: _______    Date: _______

| | |
|---|---|
| **Compensation** | *N/A* |
| **Right to Withdraw and Questions** | *Your participation in this research is completely voluntary. You may choose not to take part at all. If you decide to participate in this research, you may stop participating at any time. If you decide not to participate in this study or if you stop participating at any time, you will not be penalized or lose any benefits to which you otherwise qualify.*

*If you decide to stop taking part in the study, if you have questions, concerns, or complaints, or if you need to report an injury related to the research, please contact the investigator:*

**Placeholder for anonymity** |
| **Participant Rights** | *If you have questions about your rights as a research participant or wish to report a research-related injury, please contact:*

Removed for anonymity

___________

*For more information regarding participant rights, please visit:*
removed for anomynity___________

*This research has been reviewed according to the ... IRB procedures for research involving human subjects.* |

| | |
|---|---|
| **Statement of Consent** | *By clicking "I agree", you indicate that you are at least 18 years of age, you have normal or corrected-to-normal hearing, and you are proficient in English; you have read this consent form, or have had it read to you; your questions have been answered to your satisfaction and you voluntarily agree to participate in this research study. Please download or save a copy of this form for your records.*

*If you agree to participate, please click "I agree".* |

Figure 17: User consent application.

