# OpenReview forum: "MAGNET: A Multi-agent Framework for Finding Audio-Visual Needles by Reasoning over Multi-Video Haystacks"
_NeurIPS.cc/2025/Conference — NeurIPS 2025 poster_

### Official Review · Reviewer_3r1a · 2025-07-01

**Clarity:** 3
**Significance:** 3
**Originality:** 3
**Rating:** 5
**Confidence:** 4

**Summary:**

This work proposes a new benchmark and a corresponding framework for evaluating large multimodal models in audio-visual retrieval. This is useful when retrieving relevant audio-visual segments from large video corpora where the search is to be performed on multiple video sequences. The retrieval mechanism is based on retrieval-augmented generation system that identifies key moments across multiple videos. The results are impressive and they demonstrate substantial gains over baselines.

**Questions:**

Ablations study considers the importance of each of two modalities: audio and visual. However, you also obtain transcript in the form of encoded audio-visual captions. Transcripts are easier and smaller to work with, I would be interested to see the impact of using the transcript as the third modality.

How generalizable this work is to a different type of questions? For example, instead of asking to find videos showing how to do something, can you ask to identify all frames in which a particular behavior is present, e.g., standing and talking?

**Ethical Concerns:**

["NO or VERY MINOR ethics concerns only"]

**Final Justification:**

The issues that I was interested in were addressed in the author's comments. I hope they will include these additional results into the final manuscript as well. I found this work insightful and meaningful, and I my recommendation is to accept it.

**Limitations:**

I do not think limitations are discussed.

**Paper Formatting Concerns:**

format is OK

**Quality:**

3

**Strengths And Weaknesses:**

This appears to be first-of-its-kind benchmark consisting of 3100 audio-visual QA pairs of videos across diverse domains for searching across multiple video sequences. The task that is proposed to be solved is referred to as AVHaystacksQA - finding precise videos in a large collection of videos.

Two metrics are proposed to evaluate the performance: STEM - quantifies alignment errors between the ground-truth and predicted step sequences in multi-video audio-visual answer generation and MTGS - provides a balanced and interpretable assessment of segment-level grounding performance.

There is also a new model-agnostic, multi-agent training strategy designed to enhance model performance in identifying key segments across multiple videos.

Extensive evaluation, including evaluation by a human, indicates the usefulness and performance of the proposed work.

---

> ### Author Rebuttal · Authors · 2025-07-30
>
> We thank the reviewer for the constructive feedback. We are glad that the reviewer found that our proposed task is useful, with impressive results and first-of-its-kind benchmark; our merics provide a balanced and interpretable assessment with extensive evaluation.
>
> > Ablations study considers the importance of each of two modalities: audio and visual. However, you also obtain transcript in the form of encoded audio-visual captions. Transcripts are easier and smaller to work with, I would be interested to see the impact of using the transcript as the third modality.
>
> Below, we compare the performance of various input configurations that underline the importance of the audio-visual setting:
>
>
> ### AVHaystacks-Full results under different configurations
>
> |      Method                            | B@4              | MTGS-avg |
> |----------------------------------|------------------|----------|
> | $\textbf{MAGNET}_{\text{Qwen-2.5-Omni-ZS}}$ (V + A) | **28.49**       | **0.49** |
> | $\textbf{MAGNET}_{\text{Qwen-2.5-Omni-ZS}}$ (V + T) | 26.98           |     0.42     |
> | $\textbf{MAGNET}_{\text{Qwen-2.5-Omni-ZS}}$ (T)    | 20.46           |      0.21    |
> | $\textbf{MAGNET}_{\text{Qwen-2.5-Omni-ZS}}$ (V + A + T) | 27.90       |   0.48       |
> | $\textbf{MAGNET}_{\text{Qwen-2.5-Omni-FT}}$ (V + A) | **53.69**       | **0.79** |
> | $\textbf{MAGNET}_{\text{Qwen-2.5-Omni-FT}}$ (V + T) | 48.63           |     0.65     |
> | $\textbf{MAGNET}_{\text{Qwen-2.5-Omni-FT}}$ (T)    | 35.02           |      0.28    |
> | $\textbf{MAGNET}_{\text{Qwen-2.5-Omni-FT}}$ (V + A + T) | 52.91        |   0.76       |
>
> Experimental results demonstrate that adding text as the third modality over audio and visual does not result in improved performance (rather results in a marginal performance drop possibly because of increased input token size). However, using only the text modality results in a significant dip in performance across metrics, particularly on MTGS-avg (even with fine-tuning). This validates the importance of having audio and visual modalities for improved temporal grounding.
>
>
>
> > How generalizable this work is to a different type of questions? For example, instead of asking to find videos showing how to do something, can you ask to identify all frames in which a particular behavior is present, e.g., standing and talking?
>
> Below, we report the performance on two new prompts for $\text{MAGNET}_{\text{Qwen-2.5-Omni-FT}}$. Experimental results demonstrate the generalizability of our approach.
>
> | Prompt                    | B@4  | MTGS-avg |
> |---------------------------|------|----------|
> | "Standing and talking"    | 51.28| 0.72     |
> | "Sitting and playing guitar" | 52.76| 0.75     |
>
>
>
>
> > I do not think limitations are discussed.
>
> We discuss the limitations of our approach in the future works section and elaborate on the failure cases in Failure Cases (**Section H of appendix**). Per suggestion, we will make a separate section for limitations in the final draft and revise accordingly.

---

> > ### Comment · Reviewer_3r1a · 2025-08-05
> >
> > Thak you for providing details about using text as the third modality and the two examples that demonstrate model generality.  Interesting findings about using the text, they seem a bit counterintuitive, but I guess this is what this work is designed to show.  I do not have any further comments.

---

### Official Review · Reviewer_8Nrk · 2025-07-02

**Clarity:** 3
**Significance:** 2
**Originality:** 2
**Rating:** 4
**Confidence:** 5

**Summary:**

The paper introduces AVHaystacksQA, a new audio-visual question-answering task that requires retrieving and temporally grounding evidence scattered across multiple videos. The authors also propose a benchmark of QA pairs drawn from 500 YouTube videos spanning different topics, with each answer linked to at least two video segments. The paper also proposes its own model, MAGNET, which is based on retrieval-augmented multi-agent framework. Additionally, the paper defines two new metrics (STEM and MTGS) on their task.

**Questions:**

1. Section 3.2 simply averages the similarity scores from audio-visual features and caption features to rank videos. Did the authors explore learnable weighting schemes or late-fusion strategies, and if so, how did those affect recall and downstream QA quality?

2. The 500-video corpus is dominated by instructional YouTube content such as cooking and musical tutorials. Have authours tested MAGNET on domains that feature faster scene changes or less structured narration. How can authors maintain the generalization ability of their methods?

**Ethical Concerns:**

["NO or VERY MINOR ethics concerns only"]

**Final Justification:**

Thanks for the author's feedback, I still maintain my original decision.

**Limitations:**

1. The benchmark draws on only about 500 instructional video clips.  Such a narrowly focused and modest-sized collection makes it hard to judge whether the retrieval-plus-reasoning pipeline would scale or stay robust in more diverse settings.

2. Most QA pairs and step-by-step rationales are generated by GPT prompts with little human post-editing.  This can introduce the language patterns and biases of the source LLM into the benchmark, which may potentially inflate system performance for models that share training data or stylistic cues with GPT.

**Paper Formatting Concerns:**

No concerns.

**Quality:**

3

**Strengths And Weaknesses:**

Strengths

1. The paper proposes a novel task, which jointly tests multi-video linkage, audio-visual reasoning and long-context temporal grounding. This is very interesting.

2. The authors did thorough experiments, including enough comparisons, ablation studies and human evaluations.

3. The paper is well structured, flows logically, and makes complex ideas easy to follow.


Weaknesses
1. The size of dataset is relatively small. It only has  500 videos, while it contains some long videos, the small sample size may not be very helpful in this new task.

2. STEM and MTGS still hinge on fragile text-similarity thresholds, leaving them ill-equipped to accurately capture multimodal reasoning or grounding quality; as a result, the reported performance gains are hard to interpret with confidence.

3. The MAGNET's core reasoning is off-loaded to proprietary pre-trained models that are simply prompted, not modified.  Without an open-weight substitute or an algorithmic alternative, it is hard to tell whether the reported improvements come from the authors’ architectural ideas or just from the raw power of those closed models.

---

> ### Author Rebuttal · Authors · 2025-07-30
>
> We thank the reviewer for the constructive feedback. We are glad that the reviewer found that the paper proposes a novel and interesting task, the experiments are thorough, with enough ablations and human evaluations, and the paper is well structured.
>
>
> > The size of dataset is relatively small. It only has 500 videos, ... may not be very helpful in this new task.
>
> Our benchmark includes samples from **27 diverse categories**, totalling **103 hours of video**, with **69.58%** of questions involving >=2 video linkages, each verified by human annotators involving meticulous manual inspection (Cohen's $\kappa$ value of 0.87 when validating our samples with subject matter experts, indicating strong alignment with expert annotations). Please note that our benchmark includes approximately **20 videos per subject**, making it challenging.
> We manually validate each sample to ensure strict quality control, which is both tedious and time-consuming. Notably, ours is the first benchmark to offer:
>
> (i) Multi-video linkage
>
> (ii) Fine-grained AV reasoning
>
> (iii) Long-context annotation
>
> (iv) AV retrieval-based QA
> We believe our dataset construction pipeline lays a foundation for future work in this direction, which can extend the benchmark size.
>
>
>
> > STEM and MTGS still hinge on fragile text-similarity thresholds, ... performance gains are hard to interpret with confidence.
>
>
>
> We would like to emphasise the following:
>
> Specifically, STEM uses text similarity only as a filtering mechanism (with a tunable threshold, typically $\tau_{s}$ = 0.5) to determine plausible matches between steps, after which it evaluates errors in a structured manner, including step order violations, grounding mismatches (via **IoU** on video segments), and unmatched steps (both missing and hallucinated). Importantly, matched steps must not only pass the similarity threshold but also satisfy alignment via **Hungarian matching**, which encourages globally optimal correspondences rather than spurious local matches. Thus, the metric is **robust to partial paraphrasing and places more emphasis on alignment accuracy than lexical similarity**.
>
> Similarly, please note MTGS does not depend on any textual similarity at all. It purely evaluates grounding quality by measuring the temporal **intersection-over-union (IoU)** between predicted and ground truth intervals, only for video IDs that match. The merging of intervals before computing IoU ensures the metric captures both coverage and precision at the video level. In doing so, MTGS fairly evaluates grounding fidelity even in the presence of partial overlaps, thereby offering interpretable and grounded performance insights.
>
> The combination of these metrics provides a complementary and robust evaluation of both structural and temporal alignment in multimodal settings. Moreover, our performance gains under these metrics reflect meaningful improvements in temporal and procedural grounding, not just textual matches.
>
>
>
>
> > The MAGNET's core reasoning is off-loaded to proprietary pre-trained models ... it is hard to tell whether the reported improvements come from the authors’ architectural ideas ...
>
>
> We would like to clarify that MAGNET does not solely rely on proprietary models, and several key contributions go beyond the capabilities of any particular foundation model.
>
> (i) MAGNET incorporates explicit model modifications through LoRA-based fine-tuning of audio-visual (AV) agents. These agents are adapted within our framework to better handle multi-modal, temporally grounded reasoning tasks.
>
> (ii) MAGNET introduces novel algorithmic contributions, including **Salient Frame Selection**, ensuring attention is focused on both the right content and the right temporal window, facilitating efficient grounded question-answering. Crucially, these contributions are model-agnostic and can be integrated into a variety of AV models.
>
> To demonstrate this generality and to isolate the impact of our architectural innovations, we extensively evaluate MAGNET using open-source/non-proprietary models such as Video-SALMONN, Unified-IO2, and Qwen-2.5-Omni, under both zero-shot and LoRA fine-tuned settings. Our experiments show consistent improvements over their vanilla baselines (refer to Table 2), underscoring the effectiveness of MAGNET's design beyond the use of any proprietary models.
>
> Additionally, our Audio-Visual Retrieval-Augmented Generation (AV-RAG) module uses **open-source encoders** such as ImageBind for grounding external knowledge, further emphasising that the pipeline is built to support transparency and reproducibility.
>
> In summary, the reported performance gains are not solely attributable to the use of powerful proprietary models. Rather, they stem from general-purpose, open-source-compatible architectural innovations that demonstrably enhance AV understanding across non-proprietary settings.
>
>
>
> > ... Did the authors explore learnable weighting schemes or late-fusion strategies, and if so, how did those affect recall and downstream QA quality?
>
>
>
> We employed the following schemes for the retrieval scoring:
>
> (i) **Scheme 1**: We employ a learnable weighting parameter $\alpha$ wherein weighted average Sim_avg = $\alpha$ * S(av, q) + (1 - $\alpha$) * S(cap, q). We use the training set to learn this $\alpha$ parameter.
>
> (ii) **Scheme 2**: We compute Hadamard fusion between S(av, q) and S(cap, q) and obtain final similarity scores.
>
> (iii) **Scheme 3**: We concatenate the S(av, q) and S(cap, q) and train a linear layer (with output dimension same as S(av, q)) to obtain the final scores.
>
> As shown in the table below, even though we observe some improvements for Schemes 1 and 3, they are not substantial and involve training. On the contrary, with simple averaging, we had obtained comparable performances without involving any training. With Scheme 2, however, we have observed a drop in the retrieval and QA performances.
>
> | Retrieval | Scheme reported in paper (avg) | Scheme 1  | Scheme 2  | Scheme 3  |
> |--------|--------------------------------|-----------|-----------|-----------|
> | R@3                              | 73.15                    | 73.18 | 70.28 | 73.68 |
> | R@5                              | 79.20                    | 79.51 | 77.85 | 79.92 |
>
>
> | QA (w/ $\textbf{MAGNET}_{\text{Qwen-2.5-Omni-FT}}$ | Scheme reported in paper (avg) | Scheme 1  | Scheme 2  | Scheme 3  |
> |--------|--------------------------------|-----------|-----------|-----------|
> | B@4         | 53.69           | 53.73 | 50.36 | 54.12 |
> | GPT-Eval         | 7.56           | 7.58 | 6.81 | 7.69 |
> | H-Eval         | 4.01           | 4.01 | 3.76 | 4.08 |
> | MTGS-avg         | 0.79           | 0.80 | 0.72 | 0.82 |
>
>
>
>
> > ... Have authours tested MAGNET on domains that feature faster scene changes or less structured narration ... generalisation ability of their methods?
>
>
> Our benchmark indeed contains samples from cases that involve rapid scene changes measured with Average Scene Duration (ASD) (Cutting & Candan (2015)). Below, we report category-wise values for these scenarios:
>
> | Example category       | B@4  | Text Sim | GPT Eval | ASD (in sec) |
> |------------------------|------|----------|----------|-|
> | DIY                    | 52.17| 5.88     | 6.97     | 9.58 |
> | 3D printing            | 51.90| 4.92     | 7.12     | 7.24 |
> | Self-Defense           | 54.72| 5.23     | 8.75     | 8.19 |
> | Playing string instrument | 54.05| 5.67     | 8.17     | 12.05 |
>
> We will add these results to the final draft.
>
>
> > ... modest-sized collection makes it hard to judge whether the retrieval-plus-reasoning pipeline would scale or stay robust ...
>
>
> Please note the following practical challenges in the data curation involved in the audio-visual QA setting:
>
> (i) **Multi-Video Linkage Complexity**: Creating connections between multiple videos is inherently challenging due to the need to accurately align and synchronise content across different media. This requires sophisticated techniques to ensure that related segments are appropriately linked.
>
> (ii) **Curating Multi-Video Questions**: Formulating questions that involve multiple videos is complex because it demands a deeper understanding of the content and the relationships between the different video segments. This requires careful consideration to create meaningful and contextually relevant questions.
>
> (iii) **Precise Temporal Grounding**: Our dataset necessitates accurate temporal grounding of events across various videos, which is both resource-intensive and requires human verification to guarantee precision. This manual effort is crucial to ensure the accuracy and reliability of the linked events.
>
> Additionally, we provide a smaller subset of AVHaystacks-50 and compare performance against AVHaystacks-Full to show the scalability of our pipeline for both retrievals as well as reasoning.
>
>
> Please note, this work is the first of its kind in this direction, laying a foundation for future research in fine-grained AV retrieval and setting a precedent for similar endeavours. Future attempts can be made to expand the benchmark size and make it more comprehensive.
>
>
> > Most QA pairs and step-by-step rationales are generated by GPT prompts ... which may potentially inflate system performance ...
>
> (i) We use AV LLMs, and as shown in Tab 1, the direct application of these models does not yield meaningful performance. However, by implementing our pipeline, which includes **AV RAG**, **salient frame selection**, and **redundant token removal**, we improve over the baseline. Incorporating LoRa further enhances performance, highlighting the effectiveness of our approach.
>
> It’s important to note that models like VideoSALMONN and UnifiedIO2 are trained on public AV datasets, unlike GPT, which lacks this AV training. If stylistic biases existed, these models would perform better in a zero-shot setting without MAGNET.
>
> (ii) GPT-4 is a text-based model that only summarises text transcripts and does not directly process AV inputs.

---

### Official Review · Reviewer_A5XP · 2025-07-03

**Clarity:** 4
**Significance:** 3
**Originality:** 3
**Rating:** 5
**Confidence:** 4

**Summary:**

The authors propose a new task, benchmark dataset, and training strategy to address challenging QA and reasoning tasks on questions that require audio-visual understanding across multiple different videos. Their proposed method is the first of its kind that does this fine-grained audio-visual reasoning over multiple independent videos, and they show significant performance improvements over simple video-based RAG and single-video reasoning baselines.

**Questions:**

1. 3100 QA pairs in the proposed dataset is not very large. Some brief justification of why this scale of QA pairs is sufficient for fine-tuning and evaluating these models on this complex task should be included in the text.

2. How does the method perform on traditional single-video AV QA benchmarks? This evaluation would help confirm that the model maintains its advantage even for questions that can be answered entirely within one video. Or if this is not applicable, justification in the text for this would be important to include.

3. It is surprising that audio + visual leads to an improvement in performance in these video settings - although the questions are automatically curated to force reliance on audio comprehension, the video filtering process seems reliant on videos with spoken transcripts, which will interfere/overlap with any semantically important audio signal useful for answering the question. Was this aspect of the videos considered in the QA pair curation process?

**Ethical Concerns:**

["NO or VERY MINOR ethics concerns only"]

**Final Justification:**

The authors sufficiently address my concerns/questions, including convincing empirical results that address my concern about single-view AV datasets. The 'Accept' rating is maintained.

**Limitations:**

Yes.

**Paper Formatting Concerns:**

None.

**Quality:**

4

**Strengths And Weaknesses:**

**Quality:**
The paper is high -quality. The motivation for reasoning over multiple videos and the related work sections are comprehensive, and the method is explained formally and clearly in detail. The evaluation metrics are also well-motivated in the text.

**Clarity:**
The paper is well-written and easy to understand. The figures are clear and effective, and introduction of the task and the method in formal notation are at the appropriate level of detail.

**Significance:**
The proposed method outperforms existing baselines on several metrics by large margins, including human evaluation, LLM-as-a-judge, and their own metrics designed for this task.

**Originality:**
The work is original - they are the first to propose a method for AV QA and reasoning over collections of videos, instead of within a single video.

---

> ### Author Rebuttal · Authors · 2025-07-30
>
> We thank the reviewer for the constructive feedback. We are glad that the reviewer found that we propose a new task, benchmark dataset, and training strategy; the proposed method is the first of its kind, our approach shows significant performance improvements; the paper is high quality, comprehensive, the method is explained well and clearly, the evaluation metrics are well motivated; the paper is well written, easy to understand, with clear figures and appropriate details. The work is original and first of its kind.
>
>
> > 3100 QA pairs in the proposed dataset is not very large. Some brief justification of why this scale of QA pairs is sufficient for fine-tuning and evaluating these models on this complex task should be included in the text.
>
>
> Thanks for the question. The factors contributing to the size of our benchmark are:
>
> (i) The dataset annotation process requires manual inspection to ensure complete correctness. Although the sample curation process is automated, to ensure strict sanity we manually validate each sample, which is both tedious and time-consuming.
>
> (ii) The models evaluated in our study are already audio-visually informed, and we fine-tune them efficiently using LoRA adapters. Our salient frame selection module enhances context by focusing on key frames. With LoRA, we require fewer samples to fine-tune our audio-visual agents, as they are heavily pre-trained on audio-visual data. Importantly, we are only fine-tuning the agents, not training the entire retrieval pipeline.
>
> Please note that the SFS module can be used with a closed-source model where finetuning was not done. As seen in Tables 2 and 3 of the main paper, these methods demonstrate strong performance even without any fine-tuning.
>
> We find many works in the literature that support our claims:
>
> [a] Low-Rank Few-Shot Adaptation of Vision-Language Models - Zanella et al.
>
> [b] Does Combining Parameter-efficient Modules Improve Few-shot Transfer Accuracy? - Asadi et al.
>
> [c] Machine Translation with Large Language Models: Prompting, Few-shot Learning, and Fine-tuning with QLoRA - Zhang et al.
>
> This body of work demonstrates that employing LoRA enables these models to perform well even with limited samples. As suggested, we will add these details to the final draft.
>
>
> > How does the method perform on traditional single-video AV QA benchmarks? This evaluation would help confirm that the model maintains its advantage even for questions that can be answered entirely within one video. Or if this is not applicable, justification in the text for this would be important to include.
>
>
> Per suggestion, we are sharing the results of our proposed approach on the single video datasets LLP, LongVALE, MusicAVQA:
>
> ### Segment-level Audio-visual event parsing
> | Method   | LLP (Acc.) |
> | -------- | ---------- |
> | AVVP     | 55.4       |
> | LSLD     | 62.2       |
> | VALOR    | 66.8       |
> | MM-CSE   | 68.9       |
> | **MAGNET** | **70.1**   |
>
> ### Omni-modal temporal video grounding
> | Method        | LongVALE (mIOU) |
> |---------------|-----------------|
> | PandaGPT      | 2.2             |
> | NextGPT       | 4.0             |
> | VTimeLLM      | 6.4             |
> | LongVALE-LLM  | 11.0            |
> | **MAGNET**      | **14.6**        |
>
> ### Audio-Visual Question Answering
> | Method   | Music AVQA (Acc.) |
> |----------|-------------------|
> | AVLLM    | 45.2              |
> | OneLLM   | 47.6              |
> | AVicuna  | 49.6              |
> | **MAGNET** | **50.3**          |
>
> Please note that since these datasets do not contain multi-video linkage annotations, the experiments are carried out on single video instances; hence, the retrieval module was not employed. As can be seen, our approach demonstrates strong performance on these datasets as well, underlining its generalizability.
>
> Zero-shot performance on the above datasets indicates the generalizability of our approach.
>
>
> > It is surprising that audio + visual leads to an improvement in performance in these video settings - although the questions are automatically curated to force reliance on audio comprehension, the video filtering process seems reliant on videos with spoken transcripts, which will interfere/overlap with any semantically important audio signal useful for answering the question. Was this aspect of the videos considered in the QA pair curation process?
>
>
> Thank you for the question. We carefully select samples that rely on both visual and audio modalities when forming the QA pairs. For a detailed explanation, please refer to the demo video at the 30s mark (in the supplementary material), which addresses a key question necessary for cooking, such as “How to grip the knife.” This demonstrates the importance for a novice chef to not only **visualise** an expert's demonstration but also **hear** and understand the steps for a comprehensive learning experience, highlighting the significance of both visual and audio modalities.
>
> We request the reviewer to refer to other dataset examples in our supplementary materials, which further underscore the utility of both modalities, thereby supporting the enhanced performance of audio-visual models compared to models using only one modality.

---

### Official Review · Reviewer_HRuz · 2025-07-03

**Clarity:** 3
**Significance:** 3
**Originality:** 3
**Rating:** 4
**Confidence:** 4

**Summary:**

This Manuscript introduces AVHaystacks, a new large-scale audio-visual benchmark specifically designed to evaluate multi-video retrieval and temporal grounding capabilities of LMMs. The benchmark AVHaystacks comprises 3,100 QA pairs across a broad range of domains, where answering each query requires finding and integrating evidence from multiple videos and modalities. In conjunction, the authors propose MAGNET, a model-agnostic, multi-agent framework leveraging RAG as well as specialized audio-visual agents for improved segment localization and answer synthesis. The work also introduces two new evaluation metrics, STEM and MTGS, tailored to balanced and interpretable assessment of grounding and multi-step QA accuracy. Experimental results highlight substantial gains over strong baselines in both automatic and human assessments.

**Questions:**

Please refer to the *Strengths and Weaknesses* section.

**Ethical Concerns:**

["NO or VERY MINOR ethics concerns only"]

**Limitations:**

yes

**Quality:**

3

**Strengths And Weaknesses:**

1. The submitted manuscript is clearly written, with a well-defined motivation, making it easy to understand the overall idea of the paper.
2. The proposed new task, method, dataset, and evaluation metrics are all valuable contributions that can help advance the audio-visual research community.
3. Has the applicability of the proposed method been considered on other audio-visual temporal localization or QA datasets (eg. AVVP, LFAV, MUSIC-AVQA, etc.)? Or perhaps on some relevant visual datasets?
4. There are noticeable traces of GPT-generated writing throughout the manuscript.
5. It is recommended to include proper citations for the baseline methods listed in the experimental tables.
6. In the Related Work section, it is suggested to add a dedicated subsection on audio-visual learning or audio-visual scene understanding.

---

> ### Author Rebuttal · Authors · 2025-07-30
>
> We thank the reviewer for the constructive feedback. We are glad that the reviewer found that: our manuscript is clearly written, with a well-defined motivation, easy to understand; the proposed task, method, dataset, and evaluation metrics are all valuable contributions, that can help the community.
>
>
> > Has the applicability of the proposed method been considered on other audio-visual temporal localization or QA datasets (eg. AVVP, LFAV, MUSIC-AVQA, etc.)? Or perhaps on some relevant visual datasets?
>
>
> Per suggestion, we are sharing the results of our proposed approach on the LLP, LongVALE, AVQA datasets (without fine-tuning on the respective training splits):
>
> ### Segment level Audio-visual event parsing
> | Method   | LLP (Acc.) |
> | -------- | ---------- |
> | AVVP     | 55.4       |
> | LSLD     | 62.2       |
> | VALOR    | 66.8       |
> | MM-CSE   | 68.9       |
> | **MAGNET** | **70.1**   |
>
> ### Omni modal temporal video grounding
> | Method        | LongVALE (mIOU) |
> |---------------|-----------------|
> | PandaGPT      | 2.2             |
> | NextGPT       | 4.0             |
> | VTimeLLM      | 6.4             |
> | LongVALE-LLM  | 11.0            |
> | **MAGNET**      | **14.6**        |
>
> ### Audio-Visual Question Answering
> | Method   | Music AVQA (Acc.) |
> |----------|-------------------|
> | AVLLM    | 45.2              |
> | OneLLM   | 47.6              |
> | AVicuna  | 49.6              |
> | **MAGNET** | **50.3**          |
>
> Please note that since these datasets do not contain multi-video linkage annotations, the experiments are carried out on single video instances, hence the retrieval module was not employed. As can be seen, our approach demonstrates strong performance on these datasets as well, underlining its generalizability.
>
>
> > It is recommended to include proper citations for the baseline methods listed in the experimental tables.
>
> Thank you for pointing this out. We have updated the tables accordingly.
>
>
> > In the Related Work section, it is suggested to add a dedicated subsection on audio-visual learning or audio-visual scene understanding.
>
> Thank you for the suggestion. We are adding a subsection in related works on Audio-Visual Learning.

---

> > ### Comment · Reviewer_HRuz · 2025-08-04
> >
> > Thank you very much for your detailed response, which has addressed most of my concerns. As before, I maintain a Borderline Accept recommendation.

---

> > > ### Author Response · Authors · 2025-08-04
> > > **Thank you!**
> > >
> > > Dear Reviewer HRuz,
> > >
> > > Thank you once again for your thoughtful comments and for taking the time to review our rebuttal. We truly appreciate your efforts!
> > >
> > > If our responses have satisfactorily addressed your concerns, may we request you to kindly consider increasing your rating.
> > >
> > > We would be happy to provide any further clarification you may need.
> > >
> > > Thank you very much for your time!

---

### Note · Authors · 2025-08-13

Dear AC and Reviewers,

We sincerely thank all reviewers and the AC for their efforts during the review and rebuttal phases. We appreciate their constructive and encouraging feedback, which has significantly enhanced the clarity and refinement of this work. We are delighted that the reviewers recognize the clarity, motivation, and quality of our manuscript, as well as the significance of our contributions.


**Reviewer HRuz** appreciated that our manuscript is clearly written, with a well-defined motivation and easy-to-understand presentation. We are glad the proposed task, method, dataset, and evaluation metrics are seen as valuable contributions that will benefit the community.


**Reviewer A5XP** acknowledged that our work proposes a novel task, benchmark dataset, and training strategy. We are encouraged that our method is considered the first of its kind, showing significant performance improvements. The reviewer also found the paper to be of high quality, comprehensive, and clearly explained with well-motivated evaluation metrics and effective use of figures and details.


**Reviewer 8Nrk** highlighted that our paper proposes a novel and interesting task, with thorough experiments supported by ablations and human evaluations. We appreciate the recognition of the well-structured presentation.


Finally, **Reviewer 3r1a** emphasized the usefulness of the proposed task and the novelty of the benchmark, complimenting our metrics for providing balanced, interpretable assessments with extensive evaluation.


During the rebuttal period, we provided detailed, point-by-point responses to address the reviewers' concerns and are **pleased to have resolved** most of them. We commit to incorporating all the points discussed during the rebuttal into the final version of the paper, along with open-sourcing the source code and dataset.


We would be sincerely grateful if, upon finding our paper strong and our rebuttals satisfactory in addressing your concerns, you might kindly consider the possibility of increasing your score and kindly lending your support toward its acceptance during the discussion phase. Once again, thank you to all the reviewers and the AC for your thorough, insightful contributions towards the smooth conduct of the conference so far!



Kind regards,

Authors

---

### Decision · Program_Chairs · 2025-09-17

**Decision:**

Accept (poster)

**Comment:**

This work introduces a new task:  named AVHaystacksQA aimed at localizing AV clips relevant to a text query and synthesizing a response from these. To this end, this work proposes:
(i) a benchmark AVHaystacks, composed of 3100 QA pairs based on 500 videos comprising 27 categories.
(ii) a RAG based framework (MAGENTA) to first localize the relevant clips from top-k promising candidate videos, and, then to synthesize the response.
(iii) two metrics: to assess the localization (using IoU), and another to assess the text response (i.e., paraphrasing mismatches, multi-step response alignment, and temporal grounding).

The proposed method is also demonstrated to work well for existing single-clip AV benchmarks  (event parsing, qa, temporal video grounding), where it performs well without fine-tuning.

The reviewers unanimously appreciated the novel AV task, specifically, the multi-clip linkage and response synthesis. The new task has the potential to spur further development of benchmarks and AV methods. The strong performance of the proposed method, combined with the development of a new task and benchmark make this a valuable contribution for the community. Hence, I am inclined to recommend this paper for acceptance. I strongly urge the authors to incorporate the feedback and the experiments presented in their response in the final version.